# Identification of key genes associated with idiopathic pulmonary fibrosis and sarcopenia by bioinformatics analysis

Lanying Shen[1☯], Zihan Yi[2☯], Jiahao Liu[2☯], Yinghua Ying[2], Yue Hu[2]*

**1** Department of Geriatric Medicine, Second Affiliated Hospital of Zhejiang University School of Medicine, Hangzhou, Zhejiang, China, **2** Key Laboratory of Respiratory Disease of Zhejiang Province, Department of Respiratory and Critical Care Medicine, Second Affiliated Hospital of Zhejiang University School of Medicine, Hangzhou, Zhejiang, China

☯ These authors are contributed equally.
* huyue88@zju.edu.cn

## Abstract

### Background

Idiopathic pulmonary fibrosis (IPF) and sarcopenia significantly affect patients' quality of life. The progression and worsening of these conditions are often associated with endoplasmic reticulum (ER) stress, a key cellular stress–response mechanism. This study aimed to investigate the involvement of ER stress in cellular dysfunction in IPF and sarcopenia by identifying ER stress-related crosstalk genes (ERSRCGs).

### Methods

Differential gene expression and weighted gene co-expression network analysis (WGCNA) were used to identify ERSRCGs. Functional enrichment analyses, including the Kyoto Encyclopedia of Genes and Genomes (KEGG), Gene Ontology (GO), Gene Set Variation Analysis (GSVA), and Gene Set Enrichment Analysis (GSEA), were performed to categorize associated pathways. Least absolute shrinkage and selection operator (LASSO) regression was applied to construct diagnostic models for sarcopenia and IPF. The CIBERSORT method was used to examine immune infiltration, and GeneMANIA was used to construct the protein–protein interaction (PPI) network.

### Results

A total of 13 ERSRCGs were substantially associated with sarcopenia and IPF. GO and KEGG analyses revealed enrichment in amino acid metabolism and xenobiotic metabolism pathways. GSEA and GSVA further highlighted the involvement of these genes in multiple biological processes and signaling pathways. LASSO regression identified *CTH* and *IDI1* for *IPF,* and *FOXO1*, *CTH, HSD11B1, GSTK1,* and *SPTSSA*

**Data availability statement:** All relevant data are within the manuscript and its Supporting Information files.

**Funding:** This work was supported in part by the "Small yet Robust" Clinical Innovation Team for Interstitial Lung Disease (grant no. CXTD202501019, to Y.H.). The funders had no role in study design, data collection and analysis, decision to publish, or preparation of the manuscript.

**Competing interests:** The authors have declared that no competing interests exist.

**Abbreviations:** IPF, Idiopathic pulmonary fibrosis; ERSRCGs, endoplasmic reticulum stress-related crosstalk genes; GEO, Gene Expression Omnibus; WGCNA, Weighted Gene Co-Expression Network Analysis; KEGG, Kyoto Encyclopedia of Genes and Genomes; GO, Gene Ontology; GSVA, Gene Set Variation Analysis; GSEA, Gene Set Enrichment Analysis; PPI, protein-protein interaction; ER, endoplasmic reticulum; UPR, unfolded protein response.

for sarcopenia. Immune infiltration analysis revealed significant correlations between ERSRCGs and immune cell populations in both diseases.

## Conclusion

This study provides novel insights into the interrelated molecular pathways between sarcopenia and IPF, underscoring the potential of ERSRCGs as diagnostic biomarkers and therapeutic targets. The developed diagnostic models highlight key genes that could significantly improve the early detection and risk assessment strategies for these conditions.

## 1 Introduction

Both IPF and sarcopenia are well established as two major health threats that are progressive and for which the management is limited [1]. IPF is a chronic lung disease, which significantly impairs respiratory function and results in a marked reduction in QoL and high morbidity and mortality rates, as well as high overall costs of care. Despite the progress in treatment: antifibrotic agents including pirfenidone, nintedanib the aetiology of IPF has not been definitely established. Moreover, these treatments have not been very effective in halting the progression of diseases related to the lungs [2]. Nonetheless, older persons are at a high risk of developing sarcopenia which is strictly described as the loss of muscle mass and the reduced physical functioning that accompanies ageing, which predisposes them to mobility limitation and poor health [3,4]. Reduced activity capacity is common in patients with IPF as this is because respiratory function is impaired; this causes muscle utilisation to reduce resulting in the onset of sarcopenia. Godfrey et al revealed that reduced muscle supportive lung function is associated with a decline in overall muscle mass, implying the effect of respiratory disease in systemic muscle wellbeing [5]. Furthermore, a chronic inflammatory state is considered a potential link between IPF and sarcopenia. These patients are always found to have systemic inflammation in the case of IPF patients, it may cause inflammation not only in the lungs additionally it affects muscles resulting in metabolic changes which leads to sarcopenia [6]. Understanding the molecular mechanisms underlying these interconnected conditions is therefore critical for identifying novel therapeutic targets.

Recent academic studies have focused on exploring how crucial a role endoplasmic reticulum (ER) stress is in the onset of some chronic diseases, including but not limited to sarcopenia and IPF. The stimulus involved in the cell's response to ER stress is the unfolded protein response (UPR), which has been associated with tissue remodeling and cellular dysfunction. This connection makes it a major research interest, especially in discovering the common molecular link between these two diseases [7,8]. In regard to IPF, studies reported that ER stress enhances fibrotic processes by inducing cell death of the epithelial cells in addition to enhancing fibroblast activity that leads to the production of a tremendous amount of extracellular matrix and impaired lung function [9]. Similarly, in sarcopenia, accumulation of misfolded

proteins as a result of ER stress may lead to muscular wasting through activating inflammatory and oxidative stress pathways, and thereby leading to muscle wasting and loss of strength [10]. Nevertheless, in light of these results, the roles of ER stress-associated pathways remain only partially defined with regard to IPF and sarcopenia, pointing to a significant knowledge deficit in the corpus of current literature.

This study aimed to identify and validate ER stress-related crosstalk genes (ERSRCGs) associated with IPF and sarcopenia. Thus, we applied significant differential expression analysis and weighted gene co-expression network analysis (WGCNA), a robust method for linking gene modules to clinicopathological traits, to identify gene networks underlying molecular interactions between the two conditions [11]. This research intends to integrate large-scale gene expression data on ER stress in deriving new insights into the common molecular pathways underlying both idiopathic pulmonary fibrosis (IPF) and sarcopenia. By using the principles identified in the present author's work and based on the findings, one may expect the consequent emergence of novel therapeutic approaches and potential diagnostic markers facilitating the management of these often difficult and disabling disorders.

## 2 Materials and methods

### 2.1 Data download

Using the R package GEOquery, we obtained the IPF-related datasets GSE24206 and GSE53845 from the Gene Expression Omnibus (GEO) database [12,13]. All samples in datasets GSE24206 and GSE53845 were derived from human lung tissue. The chip platforms for GSE24206 and GSE53845 were GPL570 and GPL6480, respectively. Specific details are provided in Table 1. Dataset GSE24206 included six control group samples and 17 IPF samples, while dataset GSE53845 contained 40 IPF samples and 8 control samples. This study used datasets GSE24206 as the training set and GSE53845 as the validation set, and both datasets included IPF and control specimens. This study does not contain any studies with humanparticipants or animals performed by any of the authors.

Using the R package GEOquery, the sarcopenia-related datasets GSE8479 and GSE1428 were also obtained from the GEO database [14,15]. Both datasets were derived from human muscle tissue. The chip platforms for GSE8479 and GSE1428 were GPL2700 and GPL96, respectively (refer to Table 1 for details). Dataset GSE8479 included 26 control samples and 25 sarcopenia samples, while dataset GSE1428 contained 10 control and 12 sarcopenia samples. This study used datasets GSE8479 as the training set and GSE1428 as the validation set, and both datasets included sarcopenia and control samples.

Endoplasmic reticulum stress-related genes (ERSRGs) are compiled in the GeneCards database (https://www.genecards.org/) [16]. Complete details about human genes can be found in the GeneCards database. "ER Stress" was the search term that we employed. 2391 ERSRGs in total were obtained after just the ERSRGs for "Protein Coding" were kept. Similar to the PubMed website's ER stress "as keywords" (https://pubmed.ncbi.nlm.nih.gov/), there is published

**Table 1. GEO microarray chip information.**

|  | IPF |  | Sarcopenia |  |
|---|---|---|---|---|
| Platform | GSE24206 | GSE53845 | GSE8479 | GSE1428 |
|  | GPL570 | GPL6480 | GPL2700 | GPL96 |
| Samples in Disease group | 17 | 40 | 25 | 12 |
| Samples in Control group | 6 | 8 | 26 | 10 |
| Tissue | Lung | Lung | Muscle | Muscle |
| Species | Homo Sapiens | Homo Sapiens | Homo Sapiens | Homo Sapiens |
| Reference | PMID: 21974901 | PMID: 25217476 | PMID: 17520024 | PMID: 15687482 |

GEO, Gene Expression Omnibus; And IPF, Idiopathic Pulmonary Fibrosis.

literature [17] in the 256 ERSRGs set. Following consolidation and de-duplication, a total of 2458 ERSRGs were obtained; for more details (refer to S1 Table in S1 File for details).

## 2.2 Differential expression analysis

The samples from the IPF dataset GSE24206 were divided into an IPF and control groups. Differentially expressed genes (GEGs) were identified using the R package limma [18]. IPF-related DEGs (IPFRDEGs) were defined as adjusted $p < 0.05$ and $|logFC| > 0.0$. Genes with adjusted $p < 0.05$ and $logFC > 0.0$ were considered significantly upregulated, whereas those with adjusted $p < 0.05$ and $logFC < 0.0$ were considered downregulated. Differential expression was visualized with volcano plots generated using the R program ggplot2 and heatmaps created using the R package.

Similarly, the samples from the Sarcopenia dataset GSE8479 were divided into sarcopenia and control groups. Differential expression analysis was performed using limma, and sarcopenia-related DEGs (SRDEGs) were defined as adjusted $p < 0.05$ and $|logFC| > 0.0$. Genes with $logFC > 0.0$ and adjusted $p < 0.05$ were considered upregulated, whereas those with $logFC < 0.0$ and adjusted $p < 0.05$ were considered downregulated. Visualization was performed using volcano plots created with the R package ggplot2 and expression heat maps generated using the R package pheatmap.

## 2.3 WGCNA

WGCNA [19] is a biology of systems approach that may be utilized to determine highly covariant gene sets and characterize patterns of gene association between various samples. Furthermore, given the connectivity of the gene set and the relationship between the specified gene set and the phenotype resulting from the analysis, identify candidate biomarker genes or therapeutic targets. WGCNA was performed using the R package WGCNA [20].

To retain the top 90% of genes in the IPF dataset GSE24206, we first calculated the variance of each gene. WGCNA parameters were set as follows: minimum separation = 0.2, soft-threshold power = 9, minimum module size = 100 genes, module merging height = 0.25, and a scale-free fitting index = 0.85. Genes within each module were identified, and correlations between the control and IPF groups across modules were assessed. Genes in each module were regarded as the module signature genes. Following a screening of the modules with $r \geq 0.30$, a Venn diagram was used to intersect genes from the most significant module with IPFRDEGs. The overlapping genes were defined as IPF-related module genes (IPFRMGs).

In the sarcopenia dataset GSE8479, gene variance was calculated and the top 90% ranked by variance were retained for analysis. Parameters were set as follows: soft-threshold power = 9, minimum distance = 0.2, module merging threshold = 0.2, scale-free fitting index = 0.90, and minimum module size = 100 genes. Genes in each module were recorded, and correlations between the sarcopenia and control groups were assessed. Genes in each module were considered to be hallmark module genes. Modules with $|r| > 0.30$ were selected, and a Wayne diagram was generated by the intersection of the most crucial module's genes with SRDEGs to identify sarcopenia-related module genes (SRMGs).

An intersection analysis of IPFRMGs and SRMGs was then performed, and a Venn diagram was generated to illustrate their overlap. The resulting shared genes were defined as crosstalk genes (CGs). Subsequently, CGs were intersected with ERSRGs, and another Venn diagram was generated to identify ERSRCGs.

## 2.4 Expression difference verification and correlation analysis of ERSRCGs

To evaluate the expression differences, ERSRCGs were compared between disease and control groups in both the IPF dataset GSE24206 and the sarcopenia dataset GSE8479 using subgroup comparison plots. The Spearman method was used to analyze the associations between ERSRCGs in the two datasets. The results were visualized using igraph (v1.6.0) and graph (v2.1.0) in R. Correlation strength was classified as follows: $|r| < 0.3 = 0.3$, 0.3–0.5 = low, 0.5–0.8 = moderate, and >0.8 = high.

## 2.5 Gene ontology (GO) and pathway (KEGG) enrichment analysis

GO enrichment analysis was conducted across the biological process (BP), cell component (CC), and molecular function (MF) categories [21]. Molecular Function (MF) and CC. A popular database for keeping data on diseases, medications, biological pathways, genomes, and other topics is the Kyoto Encyclopedia of Genes and Genomes (KEGG) [22]. We performed gene ontology (GO) and pathway (KEGG) enrichment analysis of ERSRCGs using the R-package clusterProfiler (Version 4.10.0) [23]. Significance thresholds were set at $p < 0.05$ and false discovery rate (FDR, q-value) < 0.25.

## 2.6 Gene set enrichment analysis (GSEA) of IPF and sarcopenia

Gene set enrichment analysis (GSEA) evaluates the distribution of gene sets within a ranked list of genes ordered by phenotypic association [24]. In this study, genes in the IPF training set GSE24206 were divided into IPF and control groups, and GSEA was performed on all genes using the R package clusterProfiler. The parameters were as follows: seed = 2020, minimum gene set size = 10 genes, and maximum = 500 genes. The C2 gene sets were accessed using the Molecular Signatures Database (MSigDB, v2023.2.Hs) [25]. Benjamini–Hochberg (BH) correction was applied, with significance thresholds set at adjusted $p < 0.05$ and FDR value (q-value) < 0.25.

Once more, we separated the genes in the sarcopenia training set into sarcopenia and control. We used the R-package clusterProfiler to conduct a GSEA on every gene in the Sarcopenia training set. The following were the GSEA's parameters: there were at least 10 and up to 500 genes in each gene group, and the seeds were in 2020. C2 gene sets are accessed via the Molecular Signatures Database (MSigDB). All. V2023.2. Hs. GSEA symbols. BH was used as the p-value correction method, and adj. $p < 0.05$ and FDR value (q-value) < 0.25 were the screening requirements for GSEA.

## 2.7 Analysis of gene set variation (GSVA)

GSVA is a non-parametric, unsupervised method that transforms a gene expression matrix into a gene set expression matrix to evaluate pathway-level enrichment across samples [26]. Using the MSigDB c2.cp.v2023.2.Hs gene sets, GSVA was applied to the sarcopenia training set GSE8479 and the IPF training set GSE24206. Differences in pathway enrichment between the IPF and control groups, as well as between the sarcopenia and control groups, were calculated. Screening criteria were set at adjusted $p < 0.05$, with BH correction for multiple testing.

## 2.8 Construction of diagnostic models for IPF and sarcopenia

Perform LASSO (Least Absolute Shrinkage and Selection Operator) regression analysis for ERSRCGs in order to generate a diagnostic model for the IPF training set GSE24206. When applying the sanctions term (the magnitude of the lambda × slope), what is derived from the LASSO regression analysis is more generalised than overfitting; this technique has modest roots in the linear regression analysis. A variable trajectory diagram and diagnostic model diagram show the outcomes of the LASSO regression analysis. The diagnostic model of IPF was arrived at using LASSO regression analysis and the Idiopathic Pulmonary Fibrosis Model Genes (IPFMGs) which comprised ERSRCGs.

Similarly, for the sarcopenia training set GSE8479, ERSRCGs were analyzed using LASSO regression to construct a sarcopenia diagnostic model. Genes identified in this model were defined as sarcopenia model genes (SMGs). As with the IPF model, variable trajectory plots and diagnostic model diagrams were used to visualize the outcomes.

Finally, genes shared between IPFMGs and SMGs were defined as Model genes.

## 2.9 Validation of the diagnostic models of IPF and sarcopenia

We used the IPF validation set GSE53845 and Sarcopenia validation set GSE1428 to validate the IPF diagnostic model and sarcopenia diagnostic model. To ascertain the diagnostic significance of the Sarcopenia model gene (SMGs) on sarcopenia and the IPF model gene (IPFMGs) on IPF.

Nomograms were constructed using the R package rms to visualize the functional relationships between the Model genes and disease diagnosis [27]. To assess the accuracy and resolution of the LASSO-based diagnostic models, calibration curves were generated. Decision curve analysis (DCA), a technique for evaluating clinical predictive models, was performed using the R package ggDCA to evaluate the clinical utility of each model (IPFMGs in GSE24206; SMGs in GSE8479) [28]. Receiver Operating Characteristic (ROC) curves for the training sets (GSE8479 and GSE24206) were then plotted using the R package pROC, and the Area Under the Curve (AUC) was calculated to evaluate the diagnostic performance of the LASSO RiskScore. LASSO-derived risk scores (RiskScore) were calculated using the following formula:

$$RiskScore = \sum_i Coefficient\,(gene_i)^* \, mRNA\;Expression\,(gene_i)$$

### 2.10 Validation of the IPF and sarcopenia models

To investigate the predictive effect of Model Genes on disease in IPF and Sarcopenia samples, RiskScores were calculated for both the disease and normal groups based on the expressions of IPFMGs and SMGs in the training and validation sets. The ROC curves were generated using the R package pROC to compare diagnostic performance. An AUC > 0.5 indicated that gene expression favored disease occurrence, with diagnostic accuracy classified as poor (0.5–0.7), moderate (0.7–0.9), or high (0.9).

### 2.11 Immune infiltration analysis of the IPF validation set

Immune cell composition was estimated from transcriptome expression data using the CIBERSORT algorithm, which applies linear support vector regression for deconvolution [29]. The immune infiltration matrix for dataset GSE53845 was obtained, and samples with immune cell fractions greater than zero were retained. A proportion histogram was plotted to visualize immune cell distribution. The Spearman algorithm was then used to assess the correlations between immune cells, and the R package heatmap (v1.0.12) was used to construct a correlation heatmap. Associations between Model genes and immune cells were also evaluated using Spearman correlation, with results visualized as a bubble plot generated in the R package ggplot2 (v3.4.4).

### 2.12 Immune infiltration analysis of the sarcopenia validation set

The CIBERSORT method was used in conjunction with the distinctive gene matrix of immune cells to filter out the data whose immune cell enrichment fraction was larger than zero. After obtaining the precise outcomes of the matrix of immune cell infiltration in dataset GSE1428, a proportion histogram was produced for presentation. We then employed the Spearman algorithm to determine the relationship between immune cells, and the R-package heatmap (Version 1.0.12) was used to construct the correlation heatmap that presented the findings of the immune cells' association analysis. The R package ggplot2 (Version 3.4.4) was used to construct the correlation bubble map that shows the results of the relationship research between Model Genes and immune cells. The Spearman algorithm was used to determine how immune cells and model genes are related.

### 2.13 Protein–protein interaction (PPI) network

Gene lists are analyzed, gene function hypotheses are generated, and genes are prioritized for functional analysis using the GeneMANIA database (https://genemania.org/) [30]. Using the GeneMANIA website, we predicted functionally related genes of model genes with interaction relationships in the STRING database to build a protein-protein interaction network (PPI Network).

## 2.14 Regulatory network construction

The regulatory network of Model Genes and miRNA was generated, and miRNAs associated with Model Genes were retrieved from the TarBase database to examine the link between Model Genes and miRNAs [31]. Additionally, Cytoscape software was utilized to visualize the mRNA-miRNA Regulatory Network [32]. The regulation of transcription factor (TF) on Model Genes was examined by combining transcription factors that were obtained from the ChIPBase database [19] and the HTFTarget database [33]. Furthermore, Cytoscape was utilized to visualize the mRNA-TF Regulatory Network.

## 2.15 Statistical analysis

All the statistical computations and all the analytics computations of the current research were performed using the R program (Version 4.2.2). To contrast two sets of continuous variables, the p-value for normally distributed variables is determined by a separate Student T-Test if not indicated otherwise. The Wilcoxon Rank Sum Test, or Mann-Whitney U Test, was utilized to compare samples of continuous variables where the distribution was not normal. When comparing more than three groups, the Kruskal–Wallis test was employed. Actual score data was utilized for Spearman correlation analysis aimed at determining correlation coefficients between several substances. Except where stated otherwise, all tests of statistical significance used a bilateral test, and the significance level was fixed at 0.05. This study does not contain any studies with human participants or animals performed by any of the authors.

# 3 Results

## 3.1 Technology roadmap

See Fig 1.

## 3.2 Analysis of differentially expressed genes in IPF and weighted gene co-expression network analysis

The IPF training set GSE24206 was first subjected to a differential analysis using R-packet limma to identify the genes that were differentially expressed in the two data groups. Examining the variations in gene expression levels between the IPF and Control groups was the goal of this. The findings were as follows: The IPF training set GSE24206 had 3596 DEGs that satisfied the criteria of |logFC| > 0.00 and adj. p < 0.05. There were 2045 down-regulated genes (logFC < 0.00 p-value < 0.05) and 1551 up-regulated genes (logFC > 0.00 adj. p < 0.05) below this cutoff. The dataset's difference analysis results were used to construct a map of volcanoes (Fig 2A). The differentially expressed genes that were up-regulated and down-regulated were shown on a heat map (Fig 2B).

The co-expression module was then screened using WGCNA for every gene in every sample of the IPF training set GSE24206. First, the built network was more consistent with the scale-free topology when evaluating the scale-free fitting index (Fig 2C), which was computed and shown under various soft thresholds. The findings demonstrate that the least soft threshold—that is, the ideal soft threshold—that satisfies the building of a scale-free network is 9 when the index of scale-free fitting is 0.85. Based on the ideal soft threshold, a co-expression network was built, and a clustering tree was used to group and label every gene in the IPF training set GSE24206 (Fig 2D). According to the results, the genes were grouped into 19 modules upon setting the screening threshold at 0.25. These modules included MEdarkgreen, MEturquoise, MEdarkred, MEblue, MEmidnightblue, MEbrown, MEdarkgrey, MEblack, MEplum1, MEcyan, MEsienna3, MEgreenyellow, MEdarkmagenta, MEsaddlebrown, MEdarkorange, MEpaleturquoise, MEdarkolivegreen, MEgrey, MEviolet. After that, group all of the genes together and see how they relate to the combined modules (Fig 2E). Lastly, according to the phrase patterns of module genes, the connection between all 19 modules' genes and the IPF and Control groups (Fig 2F) was determined. The MEdarkred module was the most important one.

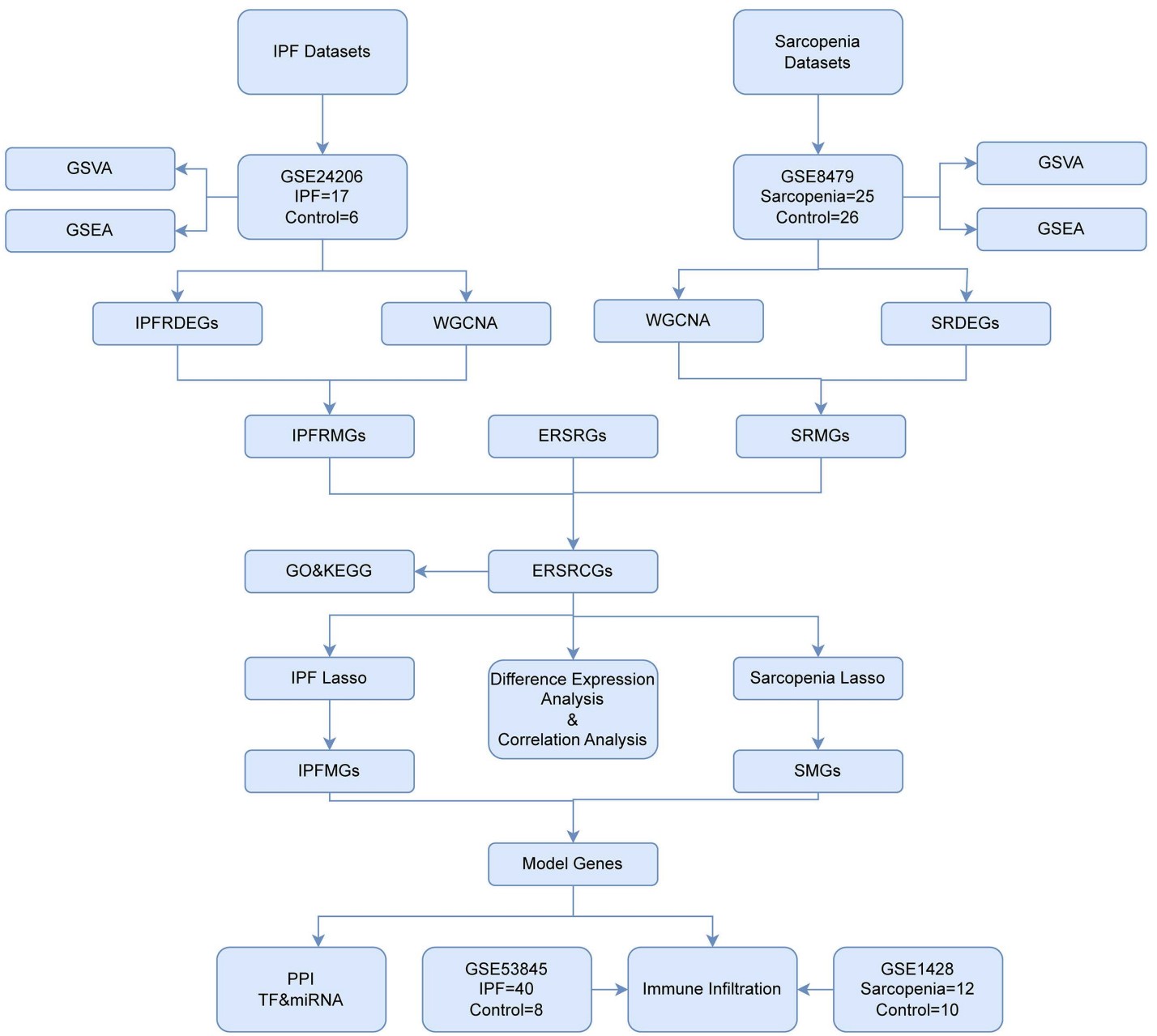

**Fig 1. Flow chart for the comprehensive analysis of IPF and sarcopenia.** IPF, Idiopathic Pulmonary Fibrosis; GSEA, Gene Set Enrichment Analysis; GSVA, Gene Set Variation Analysis; WGCNA, Weighted Correlation Network Analysis; DEGs, Differentially Expressed Genes; CGS, Crosstalk genes; IPFRDEGs, Idiopathic Pulmonary Fibrosis-Related Differentially Expressed Genes; SRDEGs, Sarcopenia-Related Differentially Expressed Genes; IPFRMGs, Idiopathic Pulmonary Fibrosis-Related Module Genes; IPfrmgs, idiopathic pulmonary fibrosis – related module genes; SRMGs, Sarcopenia-Related Module Genes; IPFMGs, IPF Model Genes; SMGs, Sarcopenia Model Genes; SMGS, sarcopenia model genes; GO, Gene Ontology; LASSO, Least Absolute Shrinkage and Selection Operator; TF, Transcription Factor; DCA, Decision Curve Analysis; ROC, Receiver Operating Characteristic; ERSRGs, Endoplasmic Reticulum Stress-Related Genes; ERSRCGs, Endoplasmic Reticulum Stress-Related Crosstalk Genes.

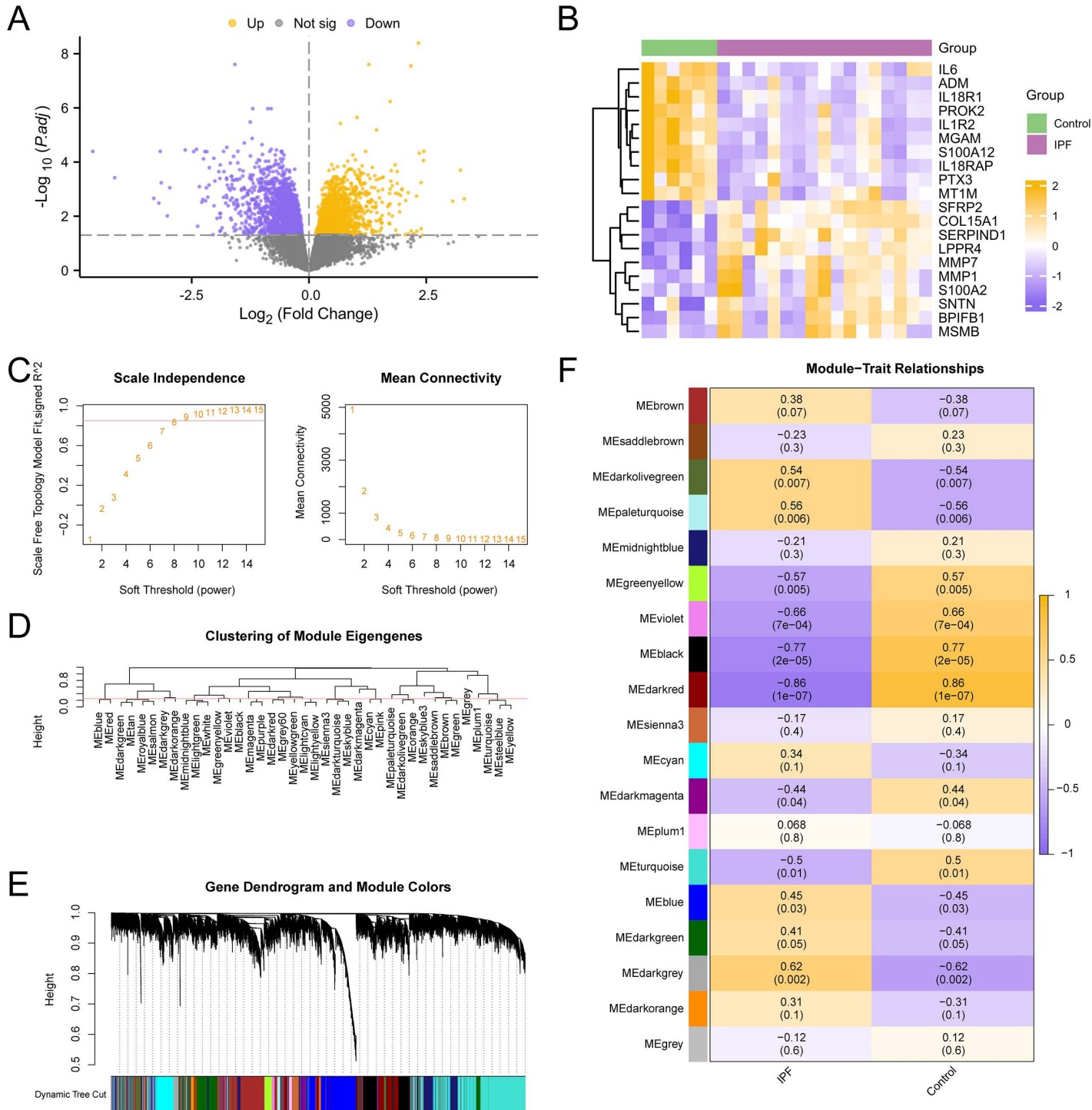

**Fig 2. Differential gene expression analysis and WGCNA for IPF datasets.** A. Differential gene expression analysis volcano map of IPF group and Control group in Idiopathic pulmonary fibrosis training concentration. B. Heat maps of expression values of top10 up-regulated and top10 down-regulated genes in the IPF training set. C. Scale-free network display of the optimal soft threshold in weighted gene co-expression network analysis (WGCNA). The left figure shows the optimal soft threshold and the right figure shows the network connectivity under different soft threshold values. D. Display of module aggregation results of all genes in the idiopathic pulmonary fibrosis training set. E. Clustering results of all genes in idiopathic pulmonary fibrosis training set were displayed. The upper part was divided into hierarchical clustering tree, and the lower part was divided into gene modules.

F. Correlation analysis results of all gene cluster modules of IPF training set GSE24206 with IPF and Control groups were presented. DEGs, Differentially Expressed Genes; WGCNA, Weighted Gene Co-Expression Network Analysis; IPFRMGs, Idiopathic Pulmonary Fibrosis-Related Module Genes. Green is the Control group and purple is the IPF group.

### 3.3 Analysis of differentially expressed genes in sarcopenia and weighted gene co-expression network analysis

The difference in the amount of gene expression between the Sarcopenia group and the Control group in the Sarcopenia training set GSE8479 was first examined utilizing the R-packet limma generated differentially expressed genes between the two groups. The results were as follows: A total of 3386 DEGs from the Sarcopenia training set GSE8479 satisfied the requirements of adj. $p < 0.05$ and $|logFC| > 0.00$. Below this cutoff, there were 1721 up-regulated genes ($logFC > 0.00$ adj. $p < 0.05$) and 1665 down-regulated genes ($logFC < 0.00$ adj. $p < 0.05$). The dataset's difference analysis results were used to construct a map of volcanoes (Fig 3A). The genes with differences in expression that were up-regulated and down-regulated were shown on a heat map (Fig 3B).

A WGCNA was then conducted on every gene in every sample of the Sarcopenia training set GSE8479 in order to screen the co-expression module of the set. First, the built-up network was made more compatible with the scale-free topology by calculating and displaying the scale-free fitting index (Fig 3C) at different soft thresholds. The results show that when the index of scale-free fitting is 0.9, the best soft threshold, or minimal soft threshold, for building the scale-free network is 9. All of the genes in the Sarcopenia training set GSE8479 were grouped and tagged with grouping information by a clustering tree, and a coexpression network was built using the ideal soft threshold (Fig 3D). The findings demonstrated that the genes were grouped in 16 modules when the screening criterion was 0.2, which were: MElightcyan, MEbrown, MEyellow, MEgreenyellow, MEblue, MEgrey, MEcyan, MEmagenta, MEpurple, MEsalmon, MEred, MEmidnightblue, MEblack, MEgreen, MEpink, MEtan. Finally, cluster all the genes and illustrate the connection of genes to the combined modules on the basis of merged modules (Fig 3E). After that, group all of the genes together and see how they relate to the combined modules (Fig 3E). Lastly, the expression profiles of the module genes were used to determine the association between all 16 modules' genes and the Sarcopenia and Control groups (Fig 3F). Lastly, MEbrown, the most important module, was tested.

### 3.4 Endoplasmic reticulum stress-related dual disease intersection gene CGs

The Wayne map was created by intersecting the genes in the MEdarkred module with a cumulative total of 3596 differentially expressed genes of idiopathic pulmonary fibrosis (IPFDEGs) (Fig 4A). A sum of 848 IPFDEGS were produced according to the Venn diagram; for more details, see S2 Table in S1 File.

The Wayne diagram was created by intersecting 3386 Sarcopenia DEGs with the genes found in the MEbrown module (Fig 4B). In all, 982 SRMGs (sarcopenia-related module genes) were identified. Refer to S3 Table in S1 File for the detailed details.

Idiopathic pulmonary fibrosis-related module Genes (IPFRMGs) were intersected with sarcopenia-related module genes (SRMGs), and the intersection Wayne map (Fig 4C) was drawn to obtain 60 Crosstalk Genes (CGs). For specific information, see S4 Table in S1 File.

Finally, the 60 Crosstalk Genes (CGs) obtained were intersected with 2458 ERSRGs and the intersection Wynn map (Fig 4D) was drawn. 13 ERSRCGs were obtained, respectively: *FOXO1, KAT2A, CTH, CTNNBIP1, HSD11B1, GABARAPL1, PKM, PHGDH, SEC31A, IDI1, GSTK1, SPTSSA, PRDM2.*

### 3.5 Expression difference verification and correlation analysis of ERSRCGs

To investigate the variations in ERSRCG expression in the IPF dataset GSE24206, 13 ERSRCGs' expression levels in the IPF group and Control group (IPF dataset GSE24206) were compared using a group comparison figure (Fig 5A). The

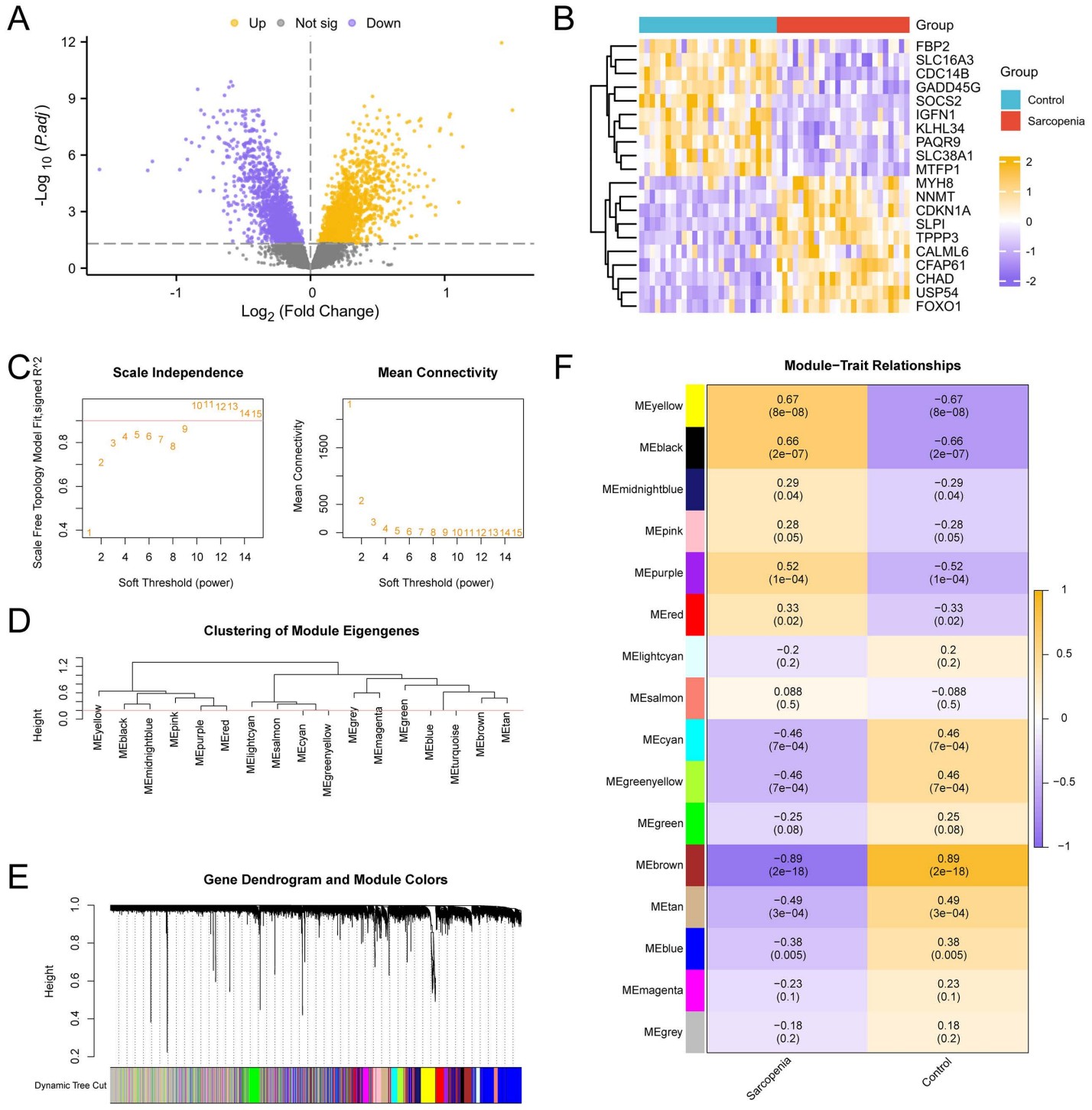

**Fig 3. Differential gene expression analysis and WGCNA for sarcopenia datasets.** A. Differential gene expression analysis Volcano map of Sarcopenia training concentration group and Control group. B. Heat maps of expression values of top10 up-regulated and top10 down-regulated genes in the sarcopenia training set. C. Scale-free network display of the best soft threshold in weighted gene co-expression network analysis (WGCNA). The left figure shows the best soft threshold, and the right figure shows the network connectivity under different soft threshold. D. Display of module aggregation results of all genes in the sarcopenia training set. E. Clustering results of all genes of the sarcopenia training set were displayed. The upper part is divided into hierarchical clustering tree, and the lower part is divided into gene modules. F. Correlation analysis results between all gene cluster

modules of the Sarcopenia training set and sarcopenia and Control groups were presented. DEGs, Differentially Expressed Genes; WGCNA, Weighted Gene Co-Expression Network Analysis; Sarcopenia, Sarcopenia; SRMGs, Sarcopenia-Related Module Genes. Blue is the Control group and red is the Sarcopenia group.

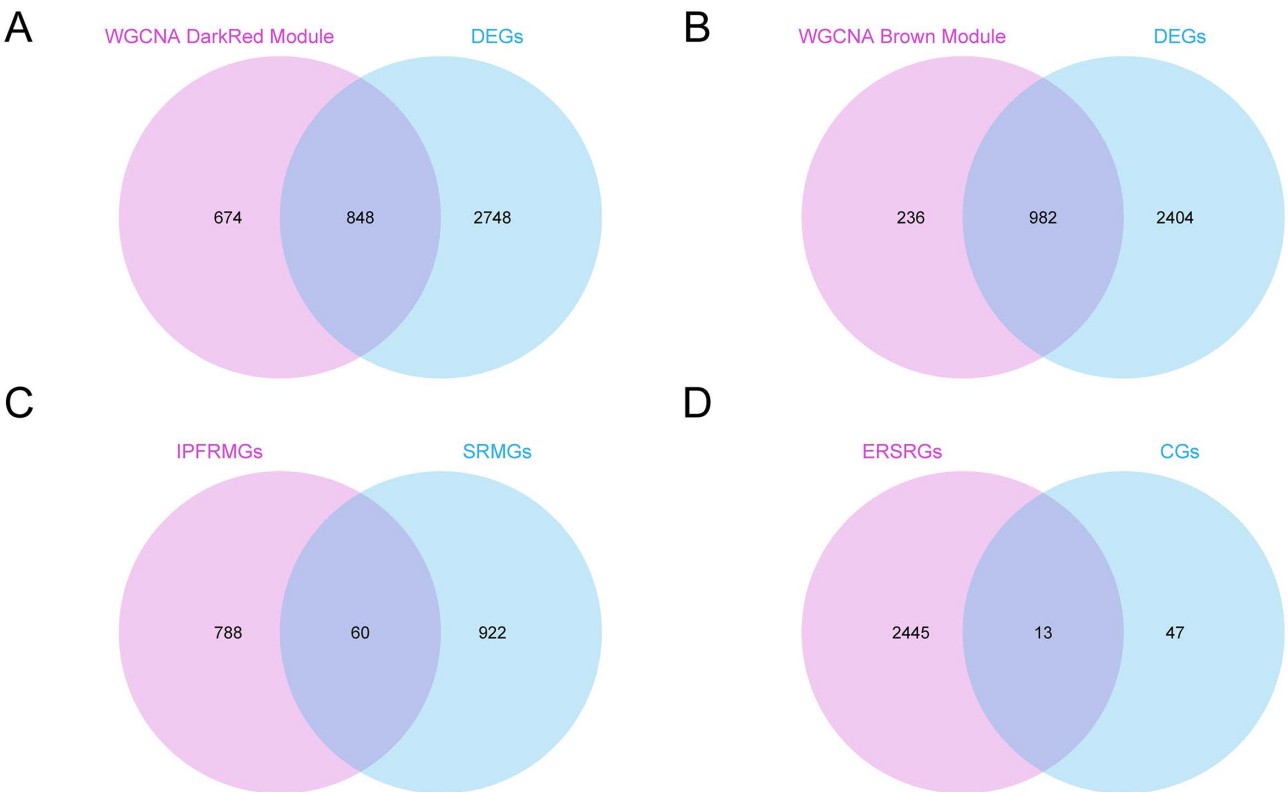

**Fig 4. Endoplasmic reticulum stress-related crosstalk genes.** A. The intersection of idiopathic pulmonary fibrosis differentially expressed genes (IPFDEGs) and module MEdarkred. Intersection diagram of B. Sarcopenia DEGs with module MEbrown. C. Intersection diagram of Idiopathic pulmonary fibrosis associated module genes (IPFRMGs) and sarcopenia associated module genes (SRMGs). D. Intersection diagram of Crosstalk Genes (CGs) and endoplasmic reticulum stress-related genes (ERSRGs). DEGs, Differentially Expressed Genes; WGCNA, Weighted Gene Co-Expression Network Analysis; IPFRMGs, Idiopathic Pulmonary Fibrosis-Related Module Genes; SRMGs, Sarcopenia-Related Module Genes; ERSRGs, Endoplasmic Reticulum Stress-Related Genes.

findings of the differential analysis revealed that the expression levels of ERSRCGs in the IPF dataset GSE24206: *CTH*, *PKM*, *IDI1*, and *SPTSSA* as well as the degrees of these genes' expression in the IPF group and Control group, were statistically significant (p-value < 0.001) (Fig 5A). In the IPF dataset GSE24206, ERSRCGs, and *PRDM2* expression levels in the IPF group and Control group were highly statistically significant (p-value < 0.01). In the IPF dataset GSE24206, the expression level of ERSRCGs *CTNNBIP1* had statistical significance (p-value < 0.05) in the Control group as well as the IPF group.

Additionally, the Group Comparison Figure (Fig 5B) showed the differences in 13 ERSRCGs levels of expression between the Sarcopenia and Control groups to investigate the variation in ERSRCGs expression in the Sarcopenia dataset GSE8479. In the Sarcopenia dataset GSE8479, the difference results showed that *FOXO1, CTH, HSD11B1, PKM, PHGDH, GSTK1, SPTSSA*, and *PRDM2* expression were highly statistically different (p-value < 0.001) between the Sarcopenia group and the Control group (Fig 5B). *KAT2A, CTNNBIP1, GABARAPL1, SEC31A*, the expression of *IDI1*

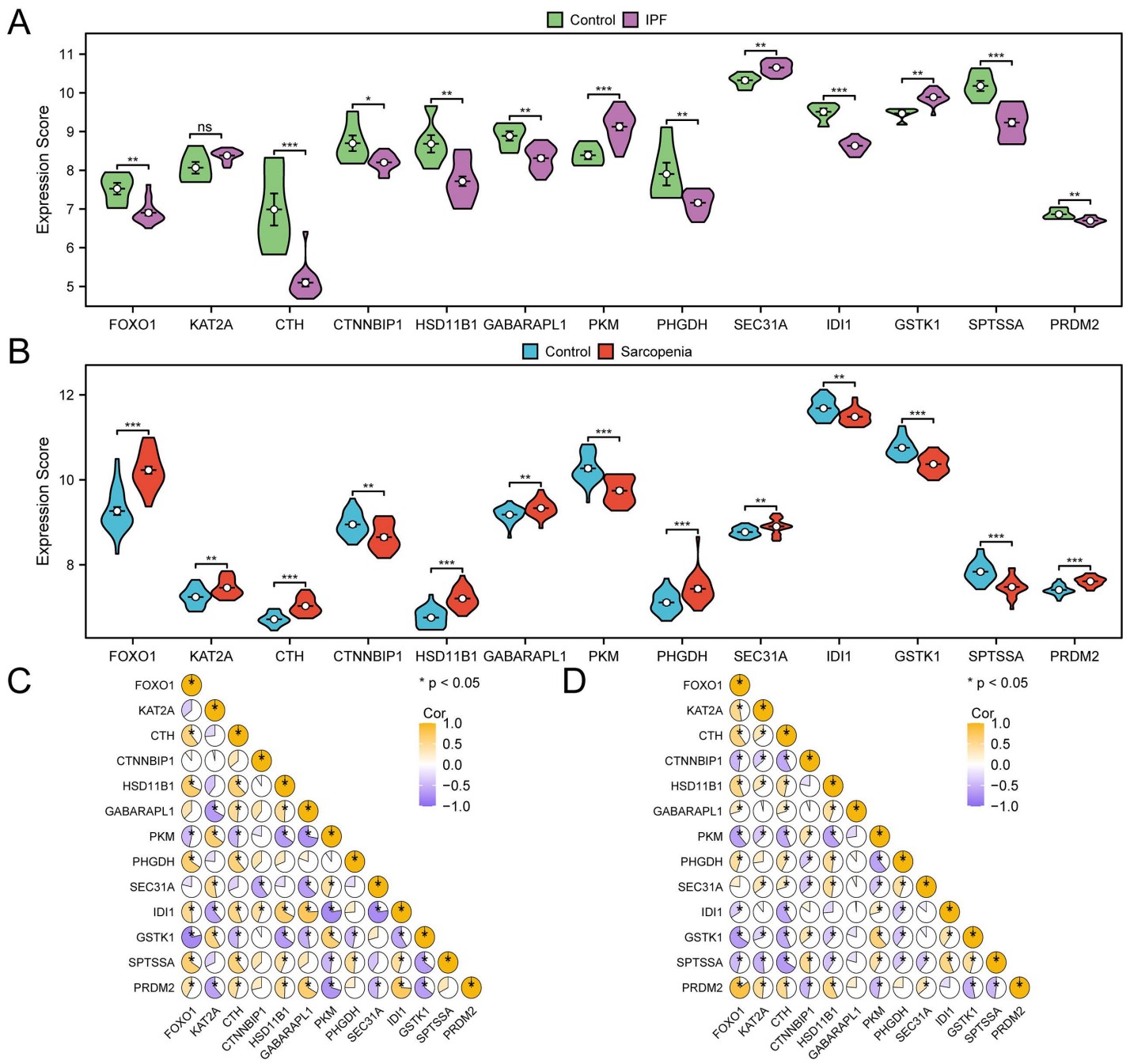

**Fig 5. Correlation analysis of ERSRCGs.** A-b. Grouping comparison of expression differences of ERSRCGs in IPF dataset GSE24206 (A) and Sarcopenia dataset GSE8479 (B). C-d. Heat maps of 13 ERSRCGs in the IPF dataset GSE24206 (C) and Sarcopenia dataset GSE8479 (D). IPF, Idiopathic Pulmonary Fibrosis; ERSRCGs, Endoplasmic Reticulum Stress-Related Crosstalk Genes. ns represented p-value > 0.05, which had no statistical significance. * meant p-value < 0.05, with statistically significant significance; ** denotes a p-value < 0.01, indicating a high level of statistical significance; *** signifies a p-value < 0.001, reflecting an exceptionally high level of statistical significance. The absolute value of the correlation coefficient (r-value) below 0.3 is weak or no correlation, between 0.3 and 0.5 is weak correlation, and between 0.5 and 0.8 is moderate correlation. In A, purple was IPF group, green was Control group. In B, red is IPF group and light blue is Control group. Yellow represents a positive association, and vine purple represents a negative association.

was highly significantly elevated (p-value < 0.01) in both the Sarcopenia and Control groups in the Sarcopenia dataset GSE8479.

We then calculated the correlation of 13 ERSRCGs in the IPF dataset GSE24206 and mapped the heat map of correlation for display (Fig 5C). The findings indicated that ERSRCGs were mainly negatively correlated with other genes.

Finally, we calculated the correlations of 13 ERSRCGs in the Sarcopenia dataset GSE8479 and drew correlation heat maps for display (Fig 5D). The findings demonstrated that the majority of the correlations between ERSRCGs and other genes were negative. The expression changes of some ERSGCs in the two diseases(IPF and Sarcopenia) were not completely consistent, suggesting disease-specific differences.

### 3.6 Gene ontology (GO) and pathway (KEGG) enrichment analysis of overlapping genes in endoplasmic reticulum stress-related dual diseases

The link of 13 ERSRCGs with IPF and Sarcopenia was further investigated by means of gene ontology (GO) and pathway (KEGG) enrichment analysis, which examined the relationship between biological processes (BP), molecular functions (MF), cellular components (CC), and biological pathways (KEGG). These 13 ERSRCGs were subjected to the gene ontology (GO) and pathway (KEGG) enrichment analysis; the particular outcomes are displayed in Table 2. As per the results, the 13 ERSRCGs were primarily enriched in the production of amino acids from the serine family, the positive regulation of gluconeogenesis, the metabolic process of glucose, the metabolic process of serine family amino acids, and other biological processes (BP); peroxisome, microbody, palmitoyltransferase complex, beta-catenin destruction complex, ATAC complex and other cellular components (CC); NAD or NADP as an acceptor, beta-catenin binding, oxidoreductase action,

**Table 2. Results of GO enrichment analysis for ERSRCGs.**

| ONTOLOGY | ID | Description | GeneRatio | BgRatio | pvalue | p.adjust | qvalue |
|---|---|---|---|---|---|---|---|
| BP | GO:0009070 | serine family amino acid biosynthetic process | 2/13 | 21/18870 | 9.13E-05 | 2.01E-02 | 1.19E-02 |
| BP | GO:0045722 | positive regulation of gluconeogenesis | 2/13 | 21/18870 | 9.13E-05 | 2.01E-02 | 1.19E-02 |
| BP | GO:0006006 | glucose metabolic process | 3/13 | 193/18870 | 2.79E-04 | 3.56E-02 | 2.11E-02 |
| BP | GO:0009069 | serine family amino acid metabolic process | 2/13 | 40/18870 | 3.37E-04 | 3.56E-02 | 2.11E-02 |
| BP | GO:0010907 | positive regulation of glucose metabolic process | 2/13 | 45/18870 | 4.27E-04 | 3.56E-02 | 2.11E-02 |
| CC | GO:0005777 | peroxisome | 2/13 | 143/19886 | 3.80E-03 | 7.02E-02 | 4.62E-02 |
| CC | GO:0042579 | microbody | 2/13 | 143/19886 | 3.80E-03 | 7.02E-02 | 4.62E-02 |
| CC | GO:0002178 | palmitoyltransferase complex | 1/13 | 11/19886 | 7.17E-03 | 7.02E-02 | 4.62E-02 |
| CC | GO:0030877 | beta-catenin destruction complex | 1/13 | 11/19886 | 7.17E-03 | 7.02E-02 | 4.62E-02 |
| CC | GO:0140672 | ATAC complex | 1/13 | 14/19886 | 9.12E-03 | 7.02E-02 | 4.62E-02 |
| MF | GO:0008013 | beta-catenin binding | 2/13 | 85/18496 | 1.58E-03 | 5.64E-02 | 1.90E-02 |
| MF | GO:0016616 | oxidoreductase activity, acting on the CH-OH group of donors, NAD or NADP as acceptor | 2/13 | 125/18496 | 3.37E-03 | 5.64E-02 | 1.90E-02 |
| MF | GO:0016614 | oxidoreductase activity, acting on CH-OH group of donors | 2/13 | 135/18496 | 3.91E-03 | 5.64E-02 | 1.90E-02 |
| MF | GO:0019903 | protein phosphatase binding | 2/13 | 142/18496 | 4.32E-03 | 5.64E-02 | 1.90E-02 |
| MF | GO:0030957 | Tat protein binding | 1/13 | 10/18496 | 7.01E-03 | 5.64E-02 | 1.90E-02 |
| KEGG | hsa01230 | Biosynthesis of amino acids | 3/13 | 75/8865 | 1.57E-04 | 7.04E-03 | 5.27E-03 |
| KEGG | hsa00260 | Glycine, serine and threonine metabolism | 2/13 | 41/8865 | 1.58E-03 | 3.55E-02 | 2.65E-02 |
| KEGG | hsa00270 | Cysteine and methionine metabolism | 2/13 | 52/8865 | 2.53E-03 | 3.79E-02 | 2.84E-02 |
| KEGG | hsa05204 | Chemical carcinogenesis – DNA adducts | 2/13 | 71/8865 | 4.66E-03 | 5.17E-02 | 3.87E-02 |
| KEGG | hsa00980 | Metabolism of xenobiotics by cytochrome P450 | 2/13 | 79/8865 | 5.74E-03 | 5.17E-02 | 3.87E-02 |

ERSRCGs, Endoplasmic Reticulum Stress-Related Crosstalk Genes; GO, Gene Ontology; BP, Biological Process; CC, Cellular Component.

and reactions with donors' CH-OH groups, protein phosphatase binding, Tat protein binding, and additional molecular functions (MF). The biosynthesis of amino acids, the metabolism of glycine, serine, threonine, cysteine, and methionine, chemical carcinogens-DNA adducts, and the cytochrome P450-mediated metabolism of xenobiotics and other biological pathways are additional areas in which it is concentrated (KEGG). A bubble diagram was used to display results from the gene ontology (GO) and pathway (KEGG) enrichment analyses (Fig 6A).

On the same token mapman, gene ontology (GO), and key gene pathway (KEGG) were employed to establish the networks of biological processes (BP), molecular function (MF), cell components (CC), and other pathways (KEGG) as depicted in the maps in (Fig 6B-E). This case should bear the matching entry and the matching molecule. As the node size increases, so do the amount of molecules in the entry.

### 3.7 Gene set enrichment analysis for IPF (GSEA)

The regulation of each gene in the IPF training set GSE24206 and the biological processes involved were examined using GSEA in order to ascertain the impact of the rate of expression of each gene on IPF. The specific results of the correlation between the impacted cell components and their molecular functions are shown in Table 3 (Fig 7A). The findings showed that the Mebarki Hcc Progenitor Wnt Up Ctnnb1 Dependent significantly enriched all of the genes in the GSE24206 training set for IPF (Fig 7B). Foroutan Integrated Tgfb Emt Dn (Fig 7E), Croonquist Il6 Deprivation Dn (Fig 7D), Stambolsky Targets Of Mutated Tp53 Dn (Fig 7C), and other physiologically significant processes and signaling pathways.

### 3.8 Gene set enrichment analysis for sarcopenia (GSEA)

To ascertain the impact of the degree of expression of every gene in the Sarcopenia training set GSE8479 on Sarcopenia, GSEA examined the biological processes involved as well as the expression of every gene in the Sarcopenia training set GSE8479. The relationship between the impacted cell components and the molecular functions carried out (Fig 8A); for particular findings refer to Table 4. The findings revealed (Fig 8B-E) that every gene in IPF training set GSE24206 was significantly enriched in Zheng Il22 Signaling Up (Fig 8B). Plasari Tgfb1 Signaling Via Nfic 10hr Dn (Fig 8C), Pid Notch Pathway (Fig 8D), Schoen Nfkb Signaling (Fig 8E) and other biologically relevant signaling pathways and functions.

### 3.9 IPF gene set variation analysis (GSVA)

To investigate the differences between the GSE24206 gene mutation analysis (GSVA) dataset and the c2. Cp. V2023.2. Hs. Symbols GMT gene set in the GSE24206 IPF and Control group, See Table 5 for specific information. Subsequently, the Top20 pathways that adj. p<0.05 were screened in descending order of absolute logFC value, and a heat chart was created to illustrate the differences in the expression of these 20 pathways in both the IPF group and the Control group (Fig 9A).

The grouping comparison diagram (Fig 9B) was then created to display the findings after the differences were confirmed using the Mann-Whitney U test. According to the results of GSVA, these pathways in the IPF group and the control group are statistically significant, please refer to Table 5 for the pathways (adj. p<0.05).

### 3.10 Sarcopenia gene set variation analysis (GSVA)

For the purpose of investigating the c2. Cp. V2023.2. Hs. Symbols. The collection of GMT genes in the data set GSE8479 less muscle disease (Sarcopenia) group and Control group, the differences between all genes of dataset GSE8479 gene mutation analysis (GSVA), for specific information, see Table 6. Then, in descending order of absolute logFC value, the top 20 pathways with adj. p<0.05 were screened. The variation in the manifestation of the 20 pathways between the Sarcopenia group and the Control group was then examined and represented using a heat map (Fig 10A).

The Mann-Whitney U test was utilized to confirm the discrepancies, and a grouping comparison diagram was used to depict the results (Fig 10B). According to the results of GSVA, these pathways in the Sarcopenia group and the control group are statistically significant, please refer to Table 6 for the pathways (adj. p<0.05).

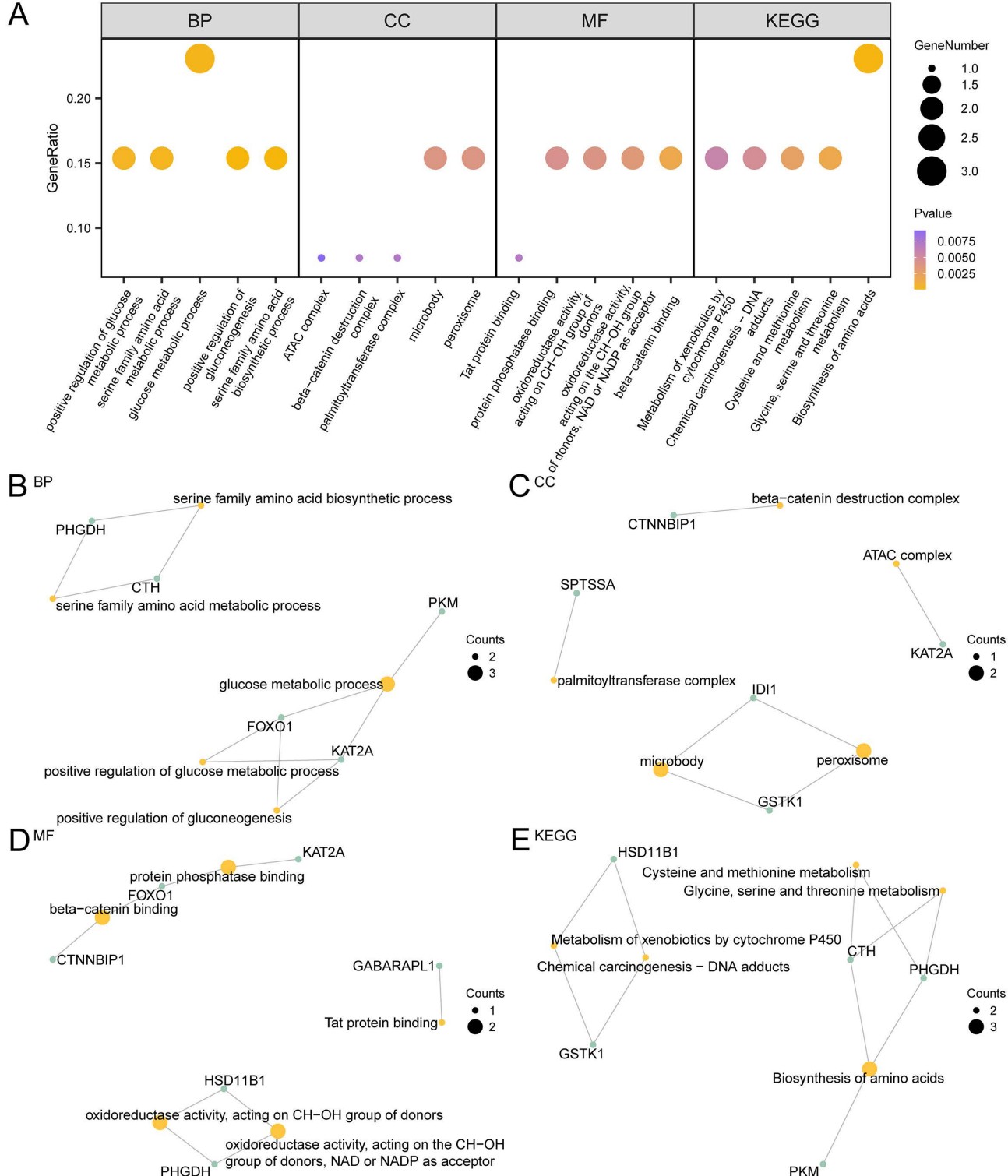

**Fig 6. GO and KEGG Enrichment Analysis for ERSRCGs.** A. Enrichment analysis results of ERSRCGs in gene ontology (GO) and pathway (KEGG) showed the bubble diagram: biological process (BP), cell component (CC), molecular function (MF) and biological pathway (KEGG). The horizontal coordinates are GO terms and KEGG terms. B-e. Gene ontology (GO) and pathway (KEGG) enrichment analysis of ERSRCGs Network diagram

showing BP (B), CC (C), MF (D) and KEGG (E). The orange nodes represent entries, the green nodes represent molecules, and the lines represent the relationships between entries and molecules. ERSRCGs, Endoplasmic Reticulum Stress-Related Crosstalk Genes; GO, Gene Ontology; KEGG, Kyoto Encyclopedia of Genes and Genomes; BP, Biological Process; CC, Cellular Component; MF, Molecular Function. The bubble size represents the number of genes in the bubble diagram, and the bubble color represents the size of the p-value value, the more yellow the p-value is, the smaller the p-value is, the more purple the p-value is, the larger the p-value is. The screening criteria for gene body (GO) and pathway (KEGG) enrichment analysis were p-value < 0.05 and FDR value (q value) < 0.25.

**Table 3. Results of GSEA for IPF datasets.**

| ID | setSize | enrichmentScore | NES | pvalue | p.adjust | qvalue |
|---|---|---|---|---|---|---|
| SENGUPTA_NASOPHARYNGEAL_CARCINOMA_DN | 299 | 0.72 | 2.91 | 1.00E-10 | 7.44E-09 | 5.52E-09 |
| CREIGHTON_ENDOCRINE_THERAPY_RESISTANCE_2 | 375 | 0.64 | 2.64 | 1.00E-10 | 7.44E-09 | 5.52E-09 |
| PID_SYNDECAN_1_PATHWAY | 46 | 0.80 | 2.52 | 1.00E-10 | 7.44E-09 | 5.52E-09 |
| NABA_COLLAGENS | 44 | 0.81 | 2.51 | 1.00E-10 | 7.44E-09 | 5.52E-09 |
| REACTOME_COLLAGEN_CHAIN_TRIMERIZATION | 44 | 0.81 | 2.51 | 1.00E-10 | 7.44E-09 | 5.52E-09 |
| REACTOME_ASSEMBLY_OF_COLLAGEN_FIBRILS_AND_OTHER_MULTIMERIC_STRUCTURES | 61 | 0.76 | 2.50 | 1.00E-10 | 7.44E-09 | 5.52E-09 |
| REACTOME_COLLAGEN_DEGRADATION | 64 | 0.73 | 2.44 | 1.00E-10 | 7.44E-09 | 5.52E-09 |
| REACTOME_COLLAGEN_FORMATION | 86 | 0.70 | 2.44 | 1.00E-10 | 7.44E-09 | 5.52E-09 |
| WP_CILIOPATHIES | 157 | 0.64 | 2.41 | 1.00E-10 | 7.44E-09 | 5.52E-09 |
| REACTOME_COLLAGEN_BIOSYNTHESIS_AND_MODIFYING_ENZYMES | 63 | 0.72 | 2.37 | 5.03E-10 | 3.22E-08 | 2.39E-08 |
| NAKAYAMA_SOFT_TISSUE_TUMORS_PCA2_UP | 82 | 0.68 | 2.36 | 1.00E-10 | 7.44E-09 | 5.52E-09 |
| RICKMAN_HEAD_AND_NECK_CANCER_D | 32 | 0.80 | 2.34 | 6.25E-09 | 3.21E-07 | 2.38E-07 |
| WP_GENES_RELATED_TO_PRIMARY_CILIUM_DEVELOPMENT_BASED_ON_CRISPR | 87 | 0.67 | 2.33 | 1.53E-10 | 1.11E-08 | 8.23E-09 |
| ANASTASSIOU_MULTICANCER_INVASIVENESS_SIGNATURE | 64 | 0.69 | 2.31 | 2.63E-09 | 1.48E-07 | 1.10E-07 |
| LEE_EARLY_T_LYMPHOCYTE_UP | 102 | 0.65 | 2.31 | 1.00E-10 | 7.44E-09 | 5.52E-09 |
| GRAHAM_NORMAL_QUIESCENT_VS_NORMAL_DIVIDING_DN | 86 | 0.66 | 2.31 | 1.04E-09 | 6.39E-08 | 4.74E-08 |
| MEBARKI_HCC_PROGENITOR_WNT_UP_CTNNB1_DEPENDENT | 78 | 0.58 | 1.98 | 1.64E-05 | 3.21E-04 | 2.38E-04 |
| STAMBOLSKY_TARGETS_OF_MUTATED_TP53_DN | 47 | 0.61 | 1.92 | 1.78E-04 | 2.40E-03 | 1.78E-03 |
| CROONQUIST_IL6_DEPRIVATION_DN | 92 | 0.54 | 1.88 | 3.77E-05 | 6.35E-04 | 4.71E-04 |
| FOROUTAN_INTEGRATED_TGFB_EMT_DN | 72 | 0.48 | 1.61 | 3.82E-03 | 2.66E-02 | 1.98E-02 |

GSEA, Gene Set Enrichment Analysis; And IPF, Idiopathic Pulmonary Fibrosis.

## 3.11 Construction of diagnostic model for IPF

First, the LASSO regression model (Fig 11A) and the LASSO variable locus diagram (Fig 11B) were created for visualization to determine the diagnostic utility of 13 ERSRCGs in IPF. According to the findings, *CTH* and *IDI1* were the two IPF model genes (IPFMGs) that were incorporated into the LASSO regression model.

In order to ascertain the 13 ERSRCGs' diagnostic significance in sarcopenia, first, the LASSO variable locus (Fig 11D) and LASSO regression model (Fig 11C) were visualized using 13 ERSRCGs. The findings demonstrated that *FOXO1, CTH, HSD11B1, GSTK1*, and *SPTSSA* were the five sarcopenia model genes (SMGs) that were included in the LASSO regression model.

## 3.12 Validation of the diagnostic model of IPF and sarcopenia

To further validate the diagnostic model of IPF, based on two idiopathic pulmonary fibrosis model genes (IPFMGs), Nomogram was created to demonstrate the relationship of IPF model genes in the IPF training set GSE24206 (Fig 12A). The

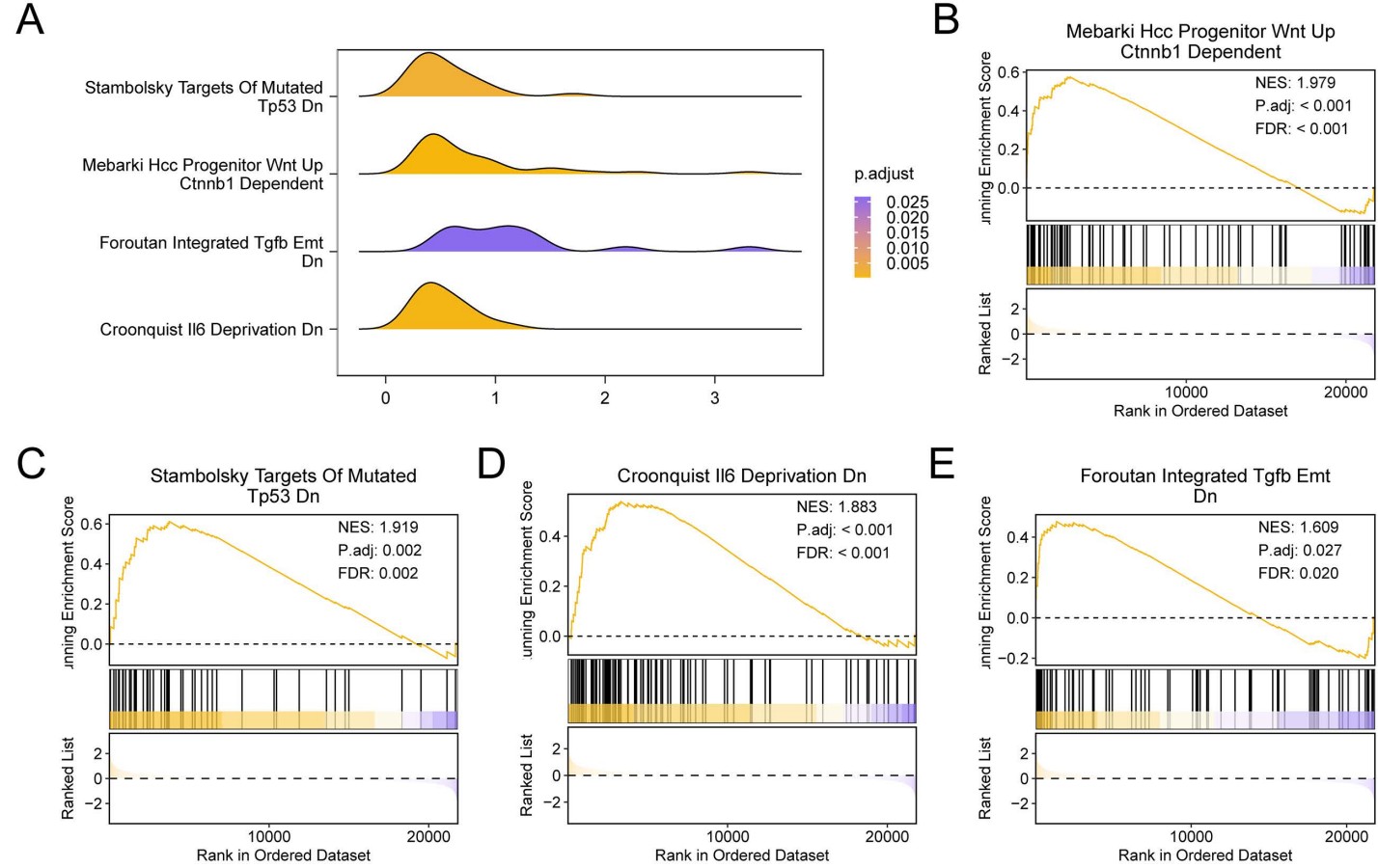

**Fig 7. GSEA for IPF datasets.** A. Gene set enrichment analysis (GSEA) of the training set of IPF shows the mountain map of 4 biological functions. B-e. Gene set enrichment analysis (GSEA) of IdiF training set showed that all genes were significantly enriched in Mebarki Hcc Progenitor Wnt Up Ctnnb1 Dependent (B), Stambolsky Targets Of Mutated Tp53 Dn (C), Croonquist Il6 Deprivation Dn (D), Foroutan Integrated Tgfb Emt Dn (E). The screening criteria for gene set enrichment analysis (GSEA) were adj. p < 0.05 and FDR value (q value) < 0.25, and the p-value correction method was Benjamini-Hochberg (BH).

findings demonstrated that, in comparison to other variables, the expression level of *IDI1* of IPFMGs had a more significant effect on the IPF diagnostic model. Compared to other variables, *CTH*'s expression level in the IPF diagnosis model was noticeably lower.

Then, using calibration analysis, a calibration curve is created to assess the IPF diagnostic model's accuracy and resolution. To fit the actual probabilities and the anticipated probability in various scenarios in order to assess the model's predictive impact on the actual outcomes (Fig 12B). The black calibration line on the diagnostic model for IPF's calibration graph is nearly aligned with the ideal model's diagonal, but it is slightly off. DCA assessed the IPF diagnosis model's clinical utility using the IPF training set GSE24206, and the findings were shown in (Fig 12C). The LASSO RiskScore is calculated using the following formula:

$$RiskScore = CTH^*(-0.145) + IDI1^*(-2.318)$$

To confirm the worth of the Sarcopenia diagnostic model, in accordance with the five Sarcopenia Model Genes, Nomogram was developed to demonstrate the link between the Sarcopenia model genes (SMGs) in the SARcopenia training

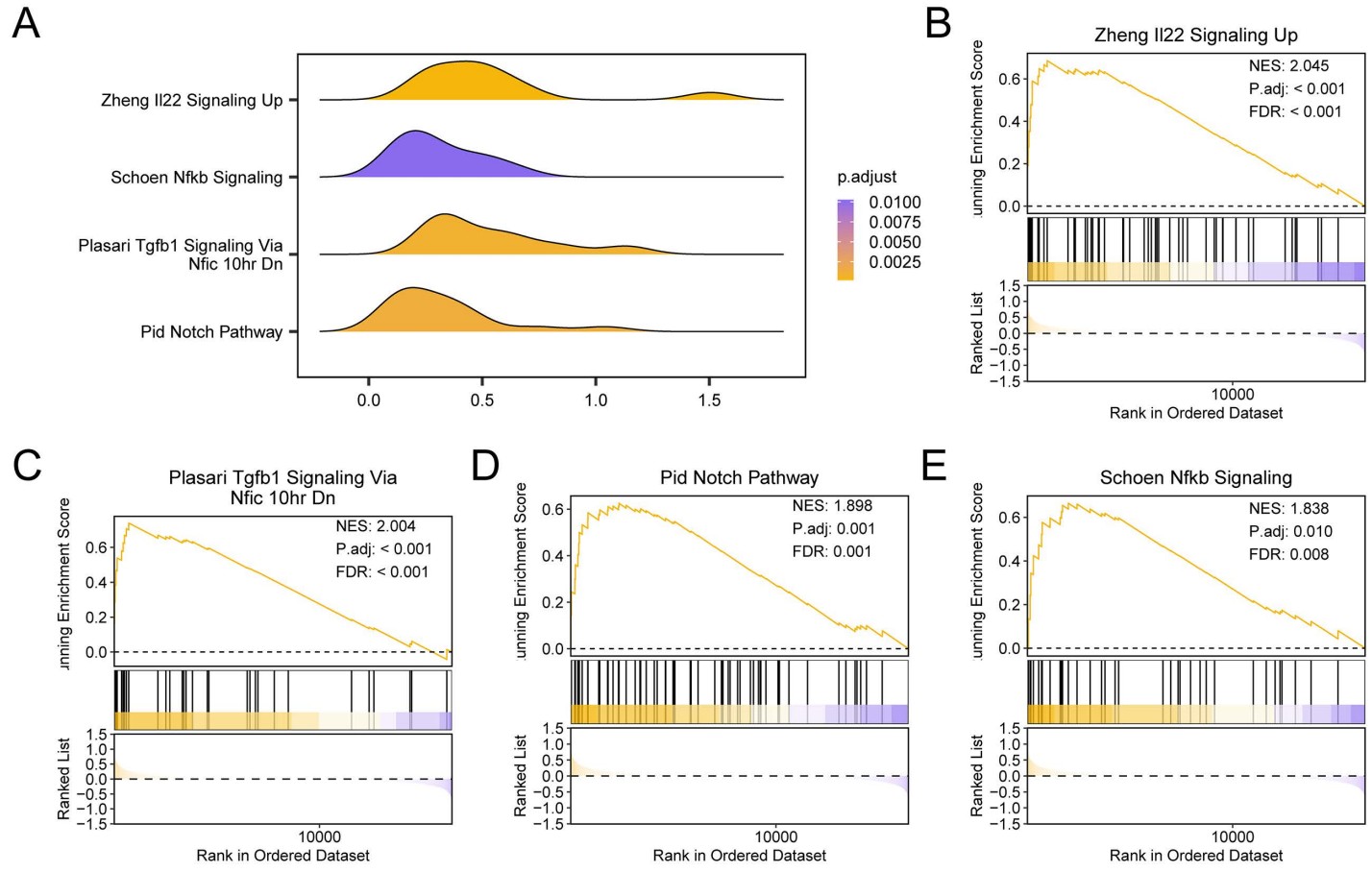

**Fig 8. GSEA for sarcopenia datasets.** A. Gene Set enrichment analysis (GSEA) of idiopathic pulmonary fibrosis training set showing 4 biological function mountain maps. Gene set enrichment analysis (GSEA) of B-E. sarcopenia training set showed significant enrichment of all genes in Zheng Il22 Signaling Up (B), Plasari Tgfb1 Signaling Via Nfic 10hr Dn (C), Pid Notch Pathway (D), Schoen Nfkb Signaling (E). The screening criteria for gene set enrichment analysis (GSEA) were adj. p < 0.05 and FDR value (q value) < 0.25, and the p-value correction method was Benjamini-Hochberg (BH).

set GSE8479 (Fig 12D). The findings demonstrated that, in comparison to other variables, the expression of the SMGs gene *GSTK1* had a substantially increased influence on the Sarcopenia diagnostic model. Compared to the different factors, the degree of *SPTSSA* expression was noticeably less useful for diagnosing sarcopenia.

The precision and clarity of the Sarcopenia diagnosis model were then evaluated by creating a calibration curve utilizing calibration analysis. To assess the predictive the model's impact on the real results according to the appropriateness of the precise probability and the model's anticipated probability in different situations (Fig 12E). The black calibration line on a calibration graph of the Sarcopenia diagnostic model is near the ideal model's diagonal, but it is a little off. DCA assessed the clinical usefulness of the Sarcopenia diagnostic model and presented the results derived from the model genes in the Sarcopenia training set GSE8479 (Fig 12F). The findings demonstrated that the model's net benefit was bigger, its effect was good, and within a certain range, its line remained constant above all positive and negative values. LASSO RiskScore is calculated by the formula that follows:

$$\text{RiskScore} = FOXO1^{*}(0.627) + CTH^{*}(1.485) + HSD11B1^{*}(2.493) + GSTK1^{*}(-1.229) + SPTSSA^{*}(-0.937)$$

**Table 4. Results of GSEA for sarcopenia datasets.**

| ID | setSize | enrichmentScore | NES | pvalue | p.adjust | qvalue |
|---|---|---|---|---|---|---|
| NAKAYAMA_SOFT_TISSUE_TUMORS_PCA1_UP | 75 | 0.80 | 2.55 | 1.00E-10 | 1.54E-08 | 1.23E-08 |
| MCLACHLAN_DENTAL_CARIES_UP | 237 | 0.67 | 2.46 | 1.00E-10 | 1.54E-08 | 1.23E-08 |
| DEMAGALHAES_AGING_UP | 53 | 0.77 | 2.29 | 8.05E-10 | 8.63E-08 | 6.92E-08 |
| JINESH_BLEBBISHIELD_TRANSFORMED_STEM_CELL_SPHERES_DN | 256 | 0.60 | 2.25 | 1.00E-10 | 1.54E-08 | 1.23E-08 |
| JINESH_BLEBBISHIELD_VS_LIVE_CONTROL_UP | 326 | 0.58 | 2.23 | 1.00E-10 | 1.54E-08 | 1.23E-08 |
| WU_CELL_MIGRATION | 173 | 0.62 | 2.23 | 1.00E-10 | 1.54E-08 | 1.23E-08 |
| ICHIBA_GRAFT_VERSUS_HOST_DISEASE_35D_UP | 135 | 0.64 | 2.21 | 1.00E-10 | 1.54E-08 | 1.23E-08 |
| CHEN_LVAD_SUPPORT_OF_FAILING_HEART_UP | 101 | 0.66 | 2.19 | 7.32E-10 | 8.13E-08 | 6.52E-08 |
| BOQUEST_STEM_CELL_UP | 248 | 0.58 | 2.16 | 1.00E-10 | 1.54E-08 | 1.23E-08 |
| FLECHNER_BIOPSY_KIDNEY_TRANSPLANT_REJECTED_VS_OK_UP | 87 | 0.66 | 2.14 | 1.54E-08 | 1.21E-06 | 9.73E-07 |
| KIM_GLIS2_TARGETS_UP | 83 | 0.66 | 2.14 | 7.01E-08 | 5.01E-06 | 4.02E-06 |
| TURASHVILI_BREAST_DUCTAL_CARCINOMA_VS_LOBULAR_NORMAL_DN | 64 | 0.69 | 2.13 | 1.36E-07 | 9.36E-06 | 7.51E-06 |
| BROWNE_HCMV_INFECTION_2HR_UP | 36 | 0.76 | 2.12 | 6.34E-07 | 3.71E-05 | 2.98E-05 |
| NADLER_OBESITY_UP | 60 | 0.69 | 2.11 | 1.47E-07 | 9.89E-06 | 7.93E-06 |
| CHIARADONNA_NEOPLASTIC_TRANSFORMATION_KRAS_CDC25_DN | 50 | 0.71 | 2.11 | 7.25E-07 | 4.19E-05 | 3.36E-05 |
| WEST_ADRENOCORTICAL_TUMOR_MARKERS_DN | 20 | 0.85 | 2.09 | 1.79E-06 | 8.86E-05 | 7.11E-05 |
| ZHENG_IL22_SIGNALING_UP | 51 | 0.69 | 2.05 | 4.45E-06 | 2.07E-04 | 1.66E-04 |
| PLASARI_TGFB1_SIGNALING_VIA_NFIC_10HR_DN | 31 | 0.74 | 2.00 | 2.55E-05 | 8.81E-04 | 7.07E-04 |
| PID_NOTCH_PATHWAY | 58 | 0.63 | 1.90 | 3.96E-05 | 1.25E-03 | 1.00E-03 |
| SCHOEN_NFKB_SIGNALING | 34 | 0.66 | 1.84 | 6.97E-04 | 1.03E-02 | 8.22E-03 |

GSEA, Gene Set Enrichment Analysis.

### 3.13 External validation of diagnostic models for IPF

The LASSO RiskScore of all samples of the validation set GSE53845 was calculated according to the LASSO RiskScore formula of the training set GSE24206 for IPF. The ROC curves of the IPF training set GSE24206 (Fig 13A) and validation set GSE53845 (Fig 13B) were then plotted using R packet pROC predicated on the LASSO risk score. In the IPF training set, GSE24206, the expression level of RiskScore demonstrated high accuracy in classifying IPF and Control groups (AUC > 0.9); in the IPF validation set, GSE53845, the expression level of RiskScore demonstrated low accuracy in classifying IPF and Control groups (0.5 < AUC < 0.7).

Similarly, the RiskScore for every sample in the validation set GSE1428 was determined using the LASSO RiskScore formula for the Sarcopenia training set GSE8479. The ROC curves of the Sarcopenia training set GSE8479 (Fig 13C) and validation set GSE1428 (Fig 13D) samples were then plotted using R packet pROC according to the LASSO risk score. The findings demonstrated that the Sarcopenia training set (GSE8479) and validation set (GSE1428) had low accuracy in categorizing Sarcopenia and Control groups based on RiskScore expression levels (0.5 < AUC < 0.7).

### 3.14 Analysis of immune infiltration in IPF

The CIBERSORT approach was utilized to determine the level of immunological infiltration of 22 different immune cell types using the data set GSE53845. According to the results of the immune infiltration investigation, a histogram showing the proportion of immune cells in dataset GSE53845 was first produced (Fig 14A). The results showed that 16 immune cell types were enriched in the IPF samples: follicular helpers, regulatory T cells, Tregs, gamma delta T cells, activated NK cells, monocytes, Macrophages M0 and M2, activated dendritic cells, resting mast cells, neutrophils, eosinophils, naive B cells, plasma cells, CD8 and CD4 T cells, and CD4 memory activated T cells.

**Table 5. Results of GSVA for IPF datasets.**

| Pathway | logFC | AveExpr | t | P.Value | adj. P.Val | B |
|---|---|---|---|---|---|---|
| KEGG MEDICUS REFERENCE CHOLESTEROL BIOSYNTHESIS | −1.15615 | −0.10017 | −5.58257 | 5.91E-07 | 0.0006 | 5.889634 |
| WP CHOLESTEROL BIOSYNTHESIS PATHWAY | −1.14626 | −0.08223 | −5.4771 | 8.81E-07 | 0.000661 | 5.525985 |
| WP CHOLESTEROL SYNTHESIS DISORDERS | −1.11176 | −0.08382 | −5.57848 | 6.00E-07 | 0.0006 | 5.875512 |
| WP NEUROINFLAMMATION | −1.07638 | −0.05732 | −5.64672 | 4.63E-07 | 0.0006 | 6.111748 |
| REACTOME REGULATION OF COMMISSURAL AXON PATHFINDING BY SLIT AND ROBO | 1.004831 | 0.075001 | 4.85426 | 8.86E-06 | 0.003353 | 3.42493 |
| KEGG MEDICUS PATHOGEN SHIGELLA IPAA TO ITGA B RHOGEF RHOA SIGNALING PATHWAY | −0.98633 | −0.07092 | −5.18215 | 2.66E-06 | 0.001595 | 4.520198 |
| REACTOME ANCHORING FIBRIL FORMATION | 0.948205 | 0.047147 | 4.851678 | 8.94E-06 | 0.003353 | 3.416414 |
| REACTOME CHOLESTEROL BIOSYNTHESIS | −0.93988 | −0.06294 | −4.89298 | 7.69E-06 | 0.003353 | 3.552862 |
| KEGG MEDICUS REFERENCE ANTIGEN PROCESSING AND PRESENTATION BY MHC CLASS II MOLECULES | 0.914578 | 0.032241 | 3.946354 | 0.000209 | 0.020273 | 0.562543 |
| KEGG MEDICUS REFERENCE MEVALONATE PATHWAY | −0.88886 | −0.02072 | −4.33344 | 5.63E-05 | 0.01401 | 1.747591 |
| WP PATHWAYS OF NUCLEIC ACID METABOLISM AND INNATE IMMUNE SENSING | 0.884769 | 0.07027 | 4.001711 | 0.000174 | 0.018414 | 0.728343 |
| REACTOME TYPE I HEMIDESMOSOME ASSEMBLY | 0.879044 | 0.091098 | 3.82089 | 0.000316 | 0.025625 | 0.191655 |
| REACTOME REGULATION OF FOXO TRANSCRIPTIONAL ACTIVITY BY ACETYLATION | −0.87645 | −0.0517 | −4.27639 | 6.85E-05 | 0.01401 | 1.569328 |
| BIOCARTA THELPER PATHWAY | 0.870226 | 0.010607 | 3.805128 | 0.000333 | 0.026078 | 0.145551 |
| WP ACQUIRED PARTIAL LIPODYSTROPHY BARRAQUER SIMONS SYNDROME | 0.85583 | 0.061561 | 4.353049 | 5.26E-05 | 0.01401 | 1.809129 |
| REACTOME RESPONSE OF EIF2AK1 HRI TO HEME DEFICIENCY | −0.85442 | −0.05692 | −4.09761 | 0.000126 | 0.01401 | 1.018563 |
| WP OMEGA 9 FATTY ACID SYNTHESIS | −0.85357 | −0.01801 | −4.10117 | 0.000125 | 0.01401 | 1.029426 |
| BIOCARTA RAB PATHWAY | −0.8525 | −0.06691 | −3.88697 | 0.000254 | 0.021812 | 0.386143 |
| WP ENTEROCYTE CHOLESTEROL METABOLISM | −0.85085 | −0.03744 | −4.48727 | 3.29E-05 | 0.010963 | 2.233972 |
| REACTOME CROSSLINKING OF COLLAGEN FIBRILS | 0.850794 | 0.027118 | 4.19339 | 9.11E-05 | 0.01401 | 1.312108 |

GSVA, Gene Set Variation Analysis; And IPF, Idiopathic Pulmonary Fibrosis.

Subsequently, correlation heat maps (Fig 14B) displayed association findings based on the quantity of 16 types of immune cell infiltration in the immune infiltration analysis in IPF samples. The majority of immune cells were shown to be significantly correlated, with resting mast cells and activated NK cells exhibiting the most positive correlation (r-value = 0.84, p-value < 0.05). Lastly, the connection between Model Genes and the quantity of immune cell infiltration was displayed using the correlation bubble diagram (Fig 14C). Most immune cells exhibited a high association, according to the correlation bubble map data. The gene CTH had the most powerful negative correlation with immune cells' T cells' gamma delta (r-value = −0.517, p-value < 0.05).

### 3.15 Analysis of immune infiltration in sarcopenia

Using the data set GSE1428, the CIBERSORT method was utilized to determine the amount of immunological infiltration of 21 different types of immune cells. The results of the immune infiltration investigation were used first to build a histogram showing the percentage of immune cells in dataset GSE1428 (Fig 15A). B cells naive, B cells memory, Plasma cells, T cells CD4 naive, T cells CD4 memory resting, T cells CD4 memory activated, T cells follicular helper, T cells regulatory Tregs, T cells gamma delta, and NK cells resting were among the 21 types of immune cell enrichment found

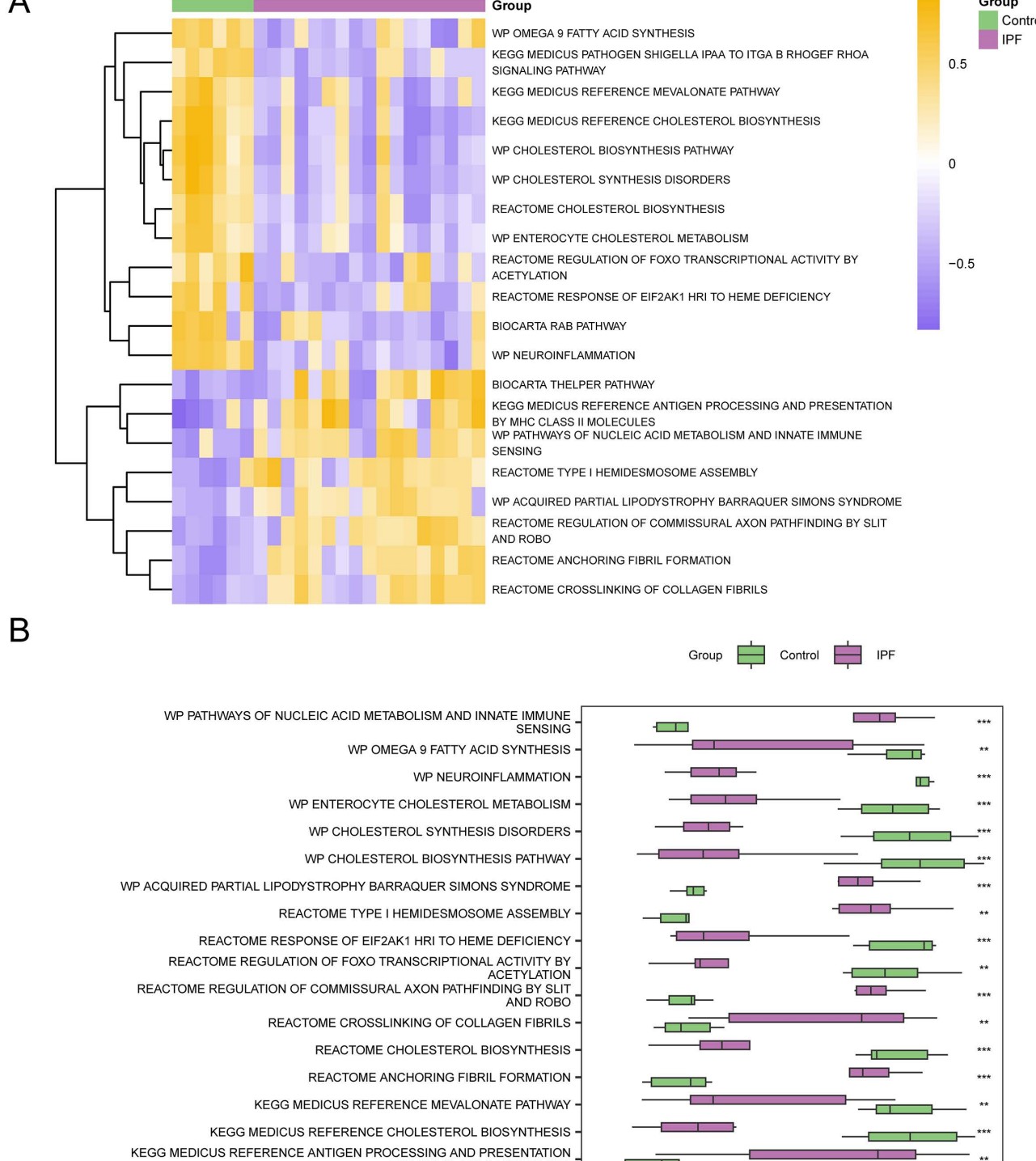

**Fig 9. GSVA Analysis.** A-B. Heat maps (A) and group comparison maps (B) of the results of gene set variation analysis (GSVA) between the IPF and Control groups in dataset GSE24206. IPF, Idiopathic Pulmonary Fibrosis; GSVA, Gene Set Variation Analysis. ** denotes a p-value < 0.01, indicating a high level of statistical significance; *** signifies a p-value < 0.001, reflecting an exceptionally high level of statistical significance. Purple represents IPF group and green represents Control group. The requirements for screening gene set variation analysis (GSVA) was adj. p < 0.05, and the p-value correction method was Benjamini-Hochberg (BH). In the heat map, violet represents low enrichment and yellow represents high enrichment.

**Table 6. Results of GSVA for sarcopenia datasets.**

| Pathway | logFC | AveExpr | t | P.Value | adj. P.Val | B |
|---|---|---|---|---|---|---|
| KEGG MEDICUS VARIANT MUTATION CAUSED ABERRANT HTT TO ELECTRON TRANSFER IN COMPLEX III | −1.33132 | −0.02997 | −10.6853 | 4.92E-16 | 2.49E-13 | 26.16409 |
| KEGG MEDICUS REFERENCE CITRATE CYCLE SECOND CARBON OXIDATION 2 | −1.32719 | −0.02163 | −8.36031 | 6.00E-12 | 6.73E-10 | 16.95717 |
| REACTOME MITOCHONDRIAL RNA DEGRADATION | −1.21164 | −0.04006 | −8.45607 | 4.04E-12 | 5.38E-10 | 17.34298 |
| WP MITOCHONDRIAL COMPLEX III ASSEMBLY | −1.21042 | −0.029 | −9.77661 | 1.85E-14 | 4.75E-12 | 22.61656 |
| REACTOME FORMATION OF ATP BY CHEMIOSMOTIC COUPLING | −1.16167 | −0.02268 | −8.39751 | 5.14E-12 | 6.00E-10 | 17.10709 |
| REACTOME CITRIC ACID CYCLE TCA CYCLE | −1.15851 | −0.0246 | −8.42105 | 4.67E-12 | 5.66E-10 | 17.20193 |
| WP TCA CYCLE AKA KREBS OR CITRIC ACID CYCLE | −1.15022 | −0.02724 | −8.12589 | 1.57E-11 | 1.59E-09 | 16.01186 |
| REACTOME CRISTAE FORMATION | −1.11263 | −0.01965 | −9.149 | 2.36E-13 | 4.48E-11 | 20.12371 |
| KEGG MEDICUS REFERENCE MITOCHONDRIAL COMPLEX UCP1 IN THERMOGENESIS | −1.10795 | −0.00134 | −11.1965 | 6.64E-17 | 7.25E-14 | 28.12069 |
| WP MITOCHONDRIAL COMPLEX IV ASSEMBLY | −1.09175 | −0.00356 | −11.0454 | 1.20E-16 | 7.25E-14 | 27.54548 |
| WP MIRNAS INVOLVED IN DNA DAMAGE RESPONSE | 1.089312 | 0.004761 | 11.09525 | 9.85E-17 | 7.25E-14 | 27.73568 |
| KEGG MEDICUS VARIANT MUTATION CAUSED ABERRANT TDP43 TO ELECTRON TRANSFER IN COMPLEX I | −1.07249 | 0.003615 | −9.35613 | 1.02E-13 | 2.20E-11 | 20.94964 |
| WP ELECTRON TRANSPORT CHAIN OXPHOS SYSTEM IN MITOCHONDRIA | −1.06573 | 0.004727 | −10.6147 | 6.51E-16 | 2.82E-13 | 25.89129 |
| KEGG MEDICUS REFERENCE ELECTRON TRANSFER IN COMPLEX I | −1.06017 | 0.014773 | −8.64354 | 1.87E-12 | 2.84E-10 | 18.09744 |
| KEGG MEDICUS VARIANT MUTATION CAUSED ABERRANT SNCA TO ELECTRON TRANSFER IN COMPLEX I | −1.05524 | 0.009165 | −8.83828 | 8.42E-13 | 1.42E-10 | 18.87966 |
| KEGG MEDICUS VARIANT MUTATION INACTIVATED PINK1 TO ELECTRON TRANSFER IN COMPLEX I | −1.04223 | 0.013883 | −8.51989 | 3.11E-12 | 4.49E-10 | 17.59998 |
| REACTOME RESPIRATORY ELECTRON TRANSPORT | −1.04216 | −5.80E-05 | −11.1197 | 8.95E-17 | 7.25E-14 | 27.82885 |
| WP OXIDATIVE PHOSPHORYLATION | −1.01476 | −0.0053 | −9.25089 | 1.56E-13 | 3.15E-11 | 20.53035 |
| WP TCA CYCLE AND DEFICIENCY OF PYRUVATE DEHYDROGENASE COMPLEX PDHC | −0.99599 | −0.04001 | −6.40509 | 1.82E-08 | 9.69E-07 | 9.115541 |
| REACTOME THE CITRIC ACID TCA CYCLE AND RESPIRATORY ELECTRON TRANSPORT | −0.99478 | −0.01176 | −11.0892 | 1.01E-16 | 7.25E-14 | 27.71248 |

GSVA, Gene Set Variation Analysis.

in the Sarcopenia cohort, according to the results. Monocytes, Dendritic cells resting, Dendritic cells activated, Mast cells resting, Mast cells activated, Macrophages M0, M1, M2, Neutrophils and Eosinophils. The difference in the expression of the abundance of immune cell infiltration within and between the Sarcopenia group and the Control group in dataset GSE1428 was subsequently displayed using subgroup comparison plots (Fig 15B).

The immune infiltration of the 21 various types of forms relative to the sarcopenia sample was then compared using correlation heat maps in the immune infiltration analysis section (Fig 15C). The results showed that a major portion of

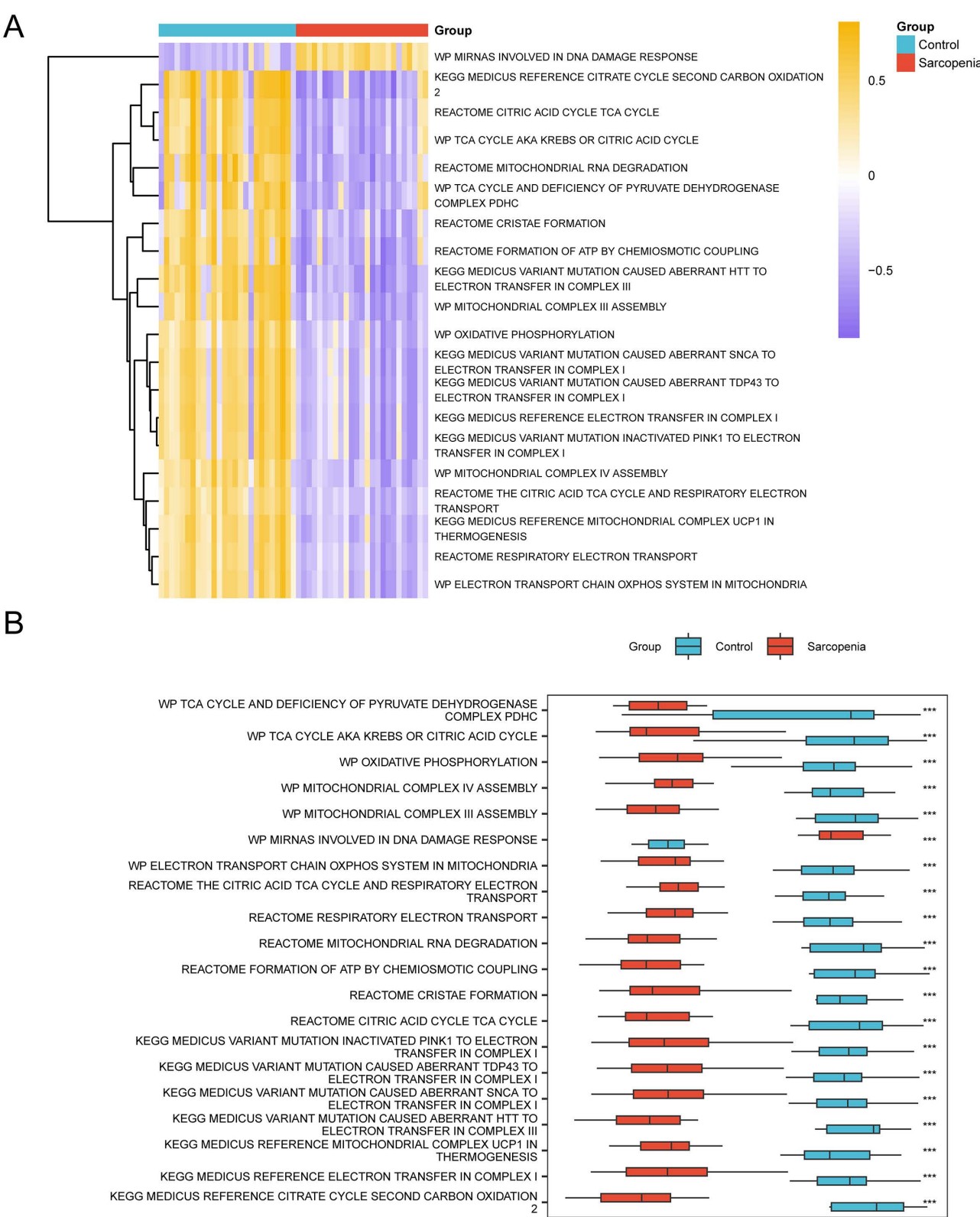

**Fig 10. GSVA Analysis.** A-B. Heat maps (A) and comparison maps (B) of the GSVA results between the Sarcopenia and Control groups in dataset GSE8479. GSVA, Gene Set Variation Analysis. *** represents p-value<0.001, which is highly statistically significant. Red represents Sarcopenia and light blue represents Control. The screening criteria for analysis of gene Set Variation (GSVA) was adj. p<0.05, and the p-value correction method was Benjamini-Hochberg (BH). In the heat map, violet represents low enrichment and yellow represents high enrichment.

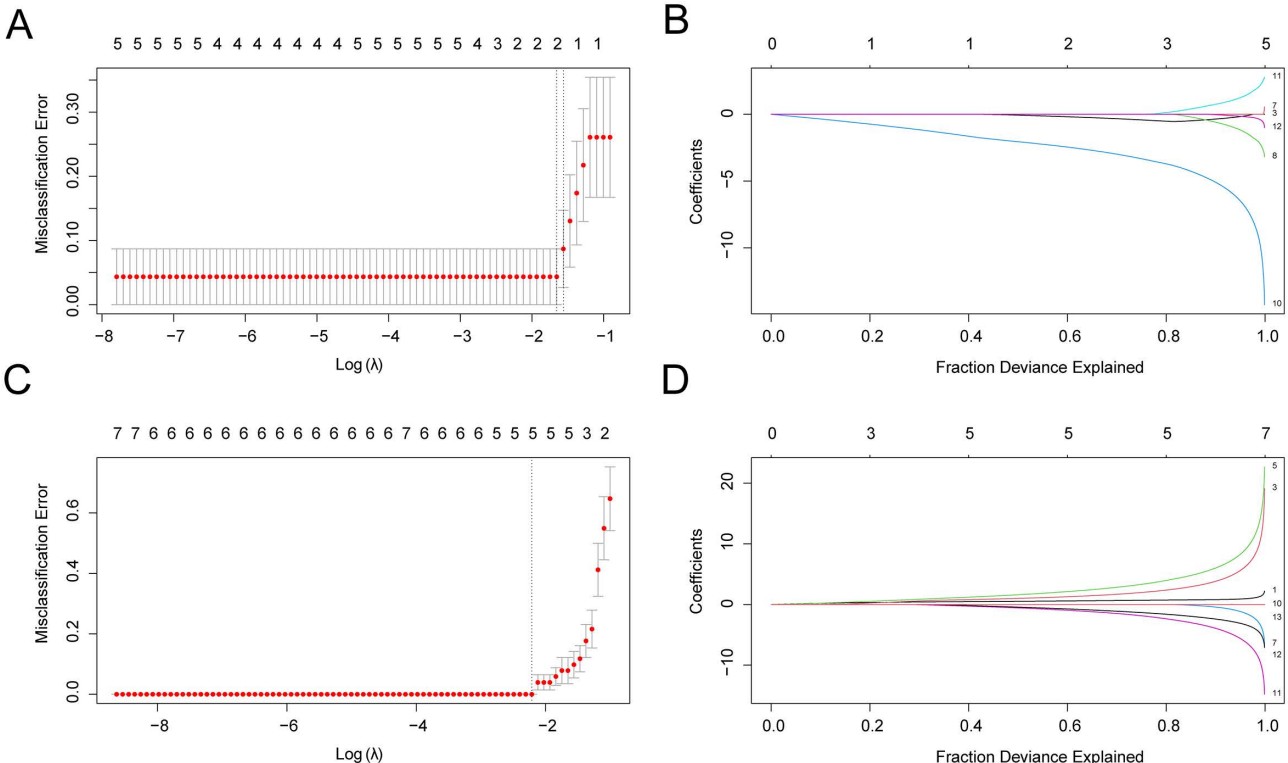

**Fig 11. Diagnostic Model of IPF and sarcopenia.** A-b. Variable locus diagram (A) and diagnostic model diagram (B) of LASSO regression model of idiopathic pulmonary fibrosis training set; C-d. Variable locus diagram (C) and diagnostic model diagram (D) of LASSO regression model for sarcopenia training set. At Least Absolute Shrinkage and Selection Operator. ERSRCGs, Endoplasmic Reticulum Stress-Related Crosstalk Genes; IPFMGs, Idiopathic Pulmonary Fibrosis Model Genes; SMGs, Sarcopenia Model Genes.

immune cells had a highly significant relationship. Out of them strongly negative association was found between resting and active mast cells (r-value=−0.796, p-value<0.05). Finally, one lollipop diagram presents the association between Model Genes and the amount of immune cell infiltration (Fig 15D). The correlation value we got on using the data from the correlation bubble diagram was found to be highly significant in most of the immune cell types and the gene *CTH* had the highest negative correlation with Eosinophils ((r-value=−0.544, p-value<0.05).

### 3.16 Construction of PPI network and regulation network of model genes

Firstly, through the GeneMANIA website, it was constructed that the interaction network (Fig 16A) of Model Genes and their functionally similar genes. The co-expression, shared protein domain, and other information between them were shown by the lines of various colors. Twenty functionally related proteins and one model gene are among them. Refer to S5 Table in S1 File for specific details.

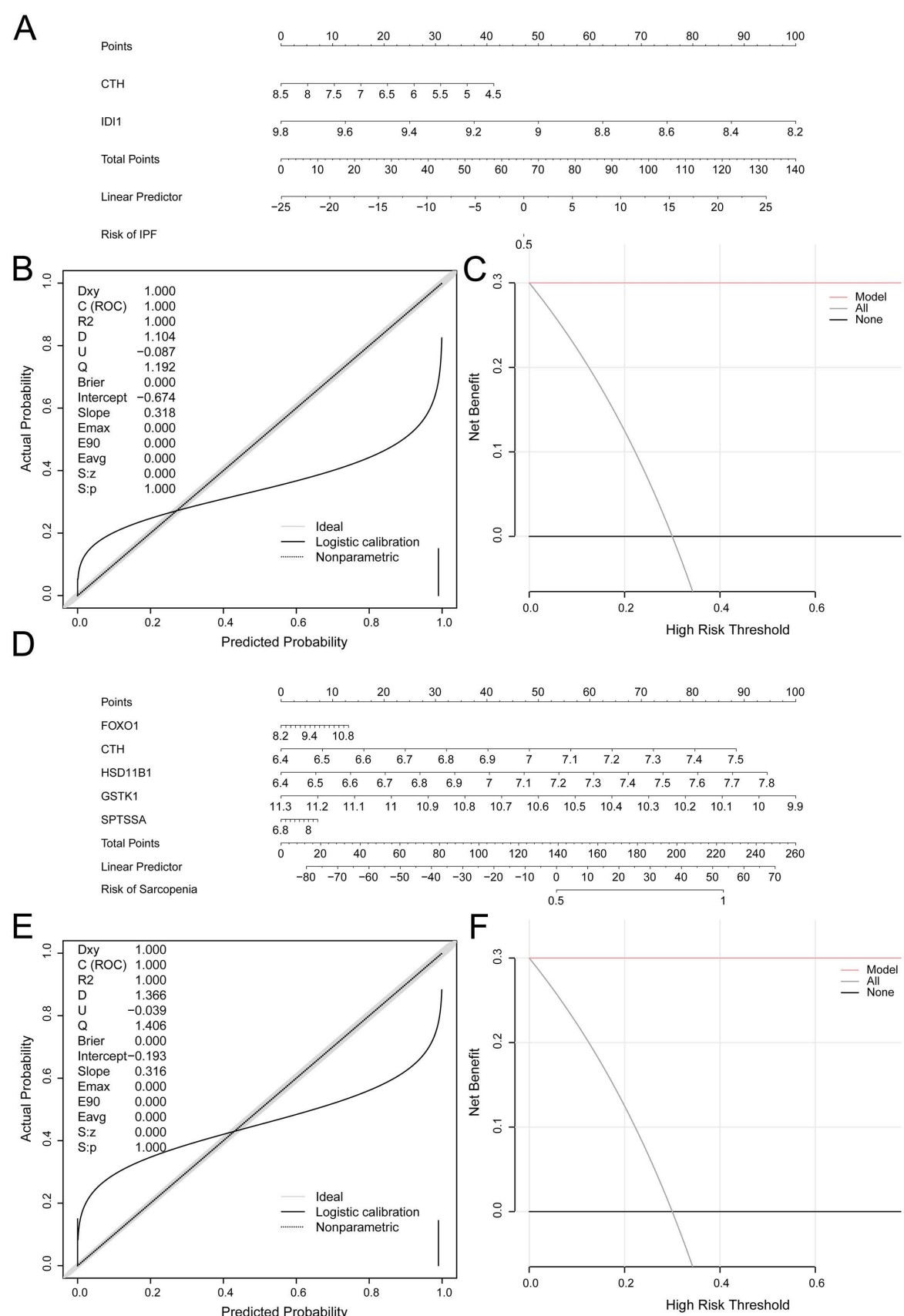

**Fig 12. Diagnostic analysis of IPF and sarcopenia.** A. Idiopathic pulmonary fibrosis model genes (IPFMGs) Nomogram of idiopathic pulmonary fibrosis training set in IPF diagnostic model. B-c. Diagnostic Model of IPF Based on Calibration graph (B) and DCA graph (C) of Model genes in the IPF training set. D. Sarcopenia model Genes (SMGs) Nomogram of the sarcopenia training set in the diagnostic model of sarcopenia. E-f. Diagnostic Model of Sarcopenia based on Calibration graph (E) and DCA graph (F) of Sarcopenia Model Genes in the sarcopenia training set. The vertical coordinate of the Calibration graph is the net income and the horizontal coordinate is the Probability Threshold or Threshold Probability. IPFMGs, Idiopathic Pulmonary Fibrosis Model Genes; SMGs, Sarcopenia Model Genes; DCA, Decision Curve Analysis; ROC Curve, Receiver Operating Characteristic Curve; ROC Curve, receiver operating characteristic curve; LASSO, at Least Absolute Shrinkage and Selection Operator.

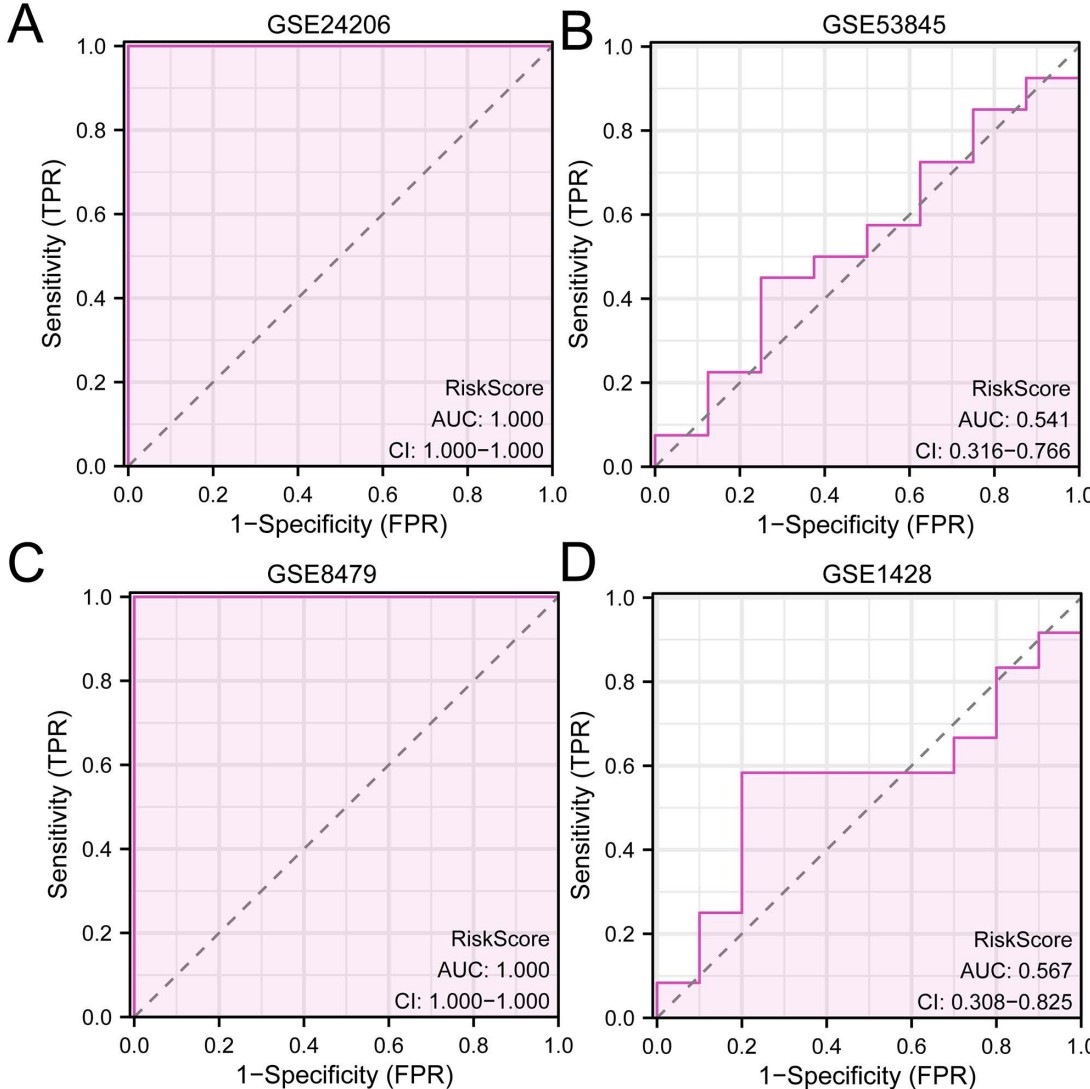

**Fig 13. ROC of the datastes.** A. ROC of the IPF training set. B. IPF validation set ROC. C. Sarcopenia training set ROC. D. Sarcopenia validation set ROC. The AUC has low accuracy at 0.5 to 0.7, and the AUC has high accuracy above 0.9.

The transcription factor (TF) binding to Model Genes was then retrieved from the ChIPBase and HTFTarget databases. To use Cytoscape software to build and show the mRNA-TF Regulatory Network (Fig 16B). They include 18 transcription factors (TFS) and 1 Model gene. For details, see S6 Table in S1 File.

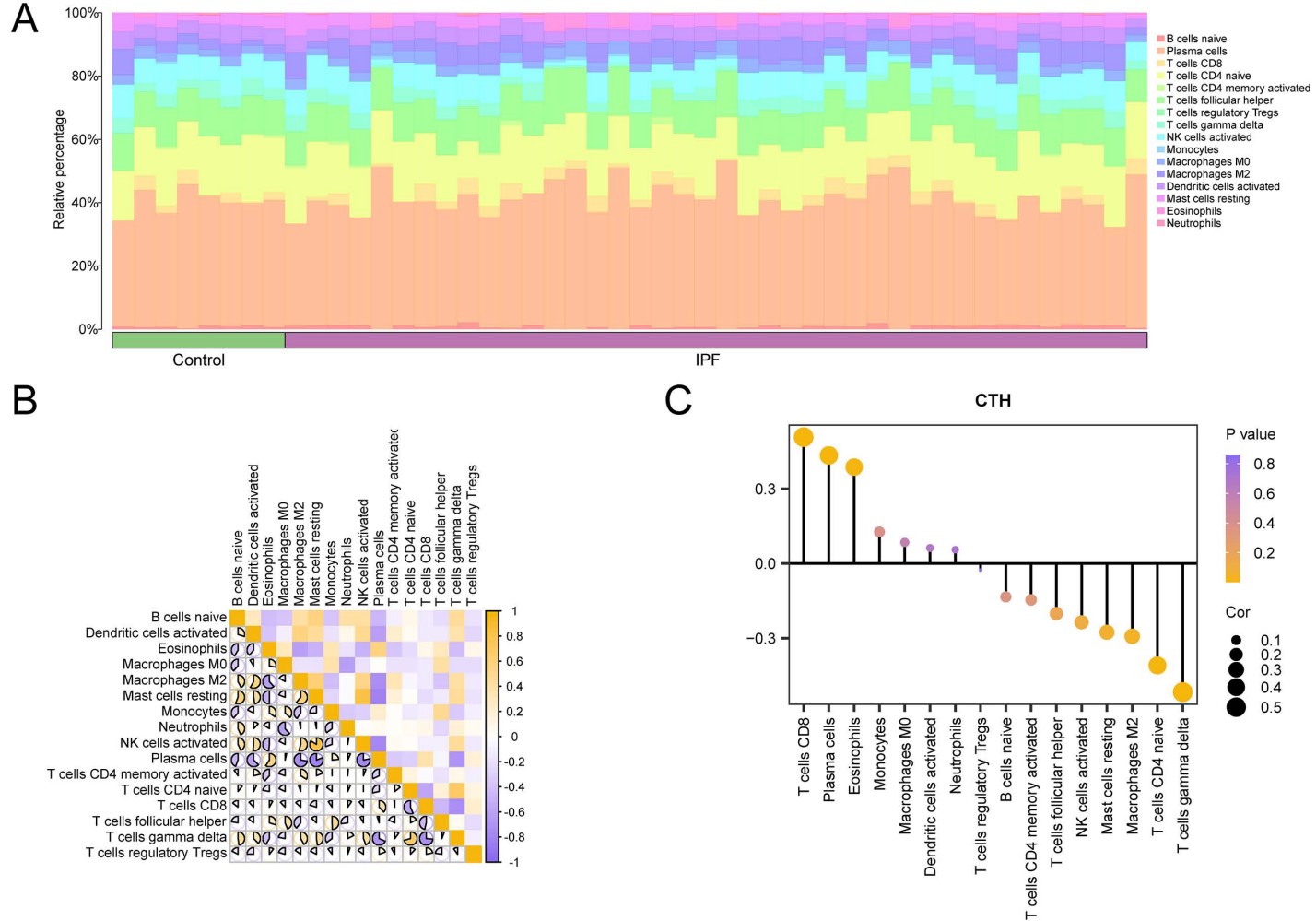

**Fig 14. Immune infiltration analysis by CIBERSORT algorithm.** A. Histogram of the proportion of immune cells in dataset GSE53845. B. Heat map of immune cell correlation in dataset GSE53845. C. Bubble map of correlation between immune cell infiltration abundance and Model Genes in dataset GSE53845. IPF, Idiopathic Pulmonary Fibrosis. The absolute value of correlation coefficient (r-value) below 0.3 is weak or no correlation, between 0.3 and 0.5 is weak correlation, between 0.5 and 0.8 is medium correlation, and above 0.8 is strong correlation. Green was the Control group, purple was the IPF group. Yellow showed positive correlation, and vine purple showed negative correlation. The depth of the color indicates the strength of the correlation.

Lastly, miRNAs associated with Model Genes were obtained from the TarBase database, and Cytoscape software was utilized to build and display the mRNA-miRNA Regulatory Network (Fig 16C). Among them, 1 Model Gene and 11 miRNAs are included. For details, see S7 Table in S1 File.

## 4 Discussion

IPF and sarcopenia are the two conditions of interest in the current research and these are two diseases that have a great impact on patients' health and quality of life. IPF is a cumulative and progressive lung ailment characterized by the accumulation of scar tissue, and probably lethal averaging a patient survival of three to five years from the commencement of the disease. The breathing is considerably impaired due to this condition, and the prognosis for the afflicted person is extremely poor [34]. Sarcopenia on the other hand is the physiological process of tissue and muscle wasting that occurs

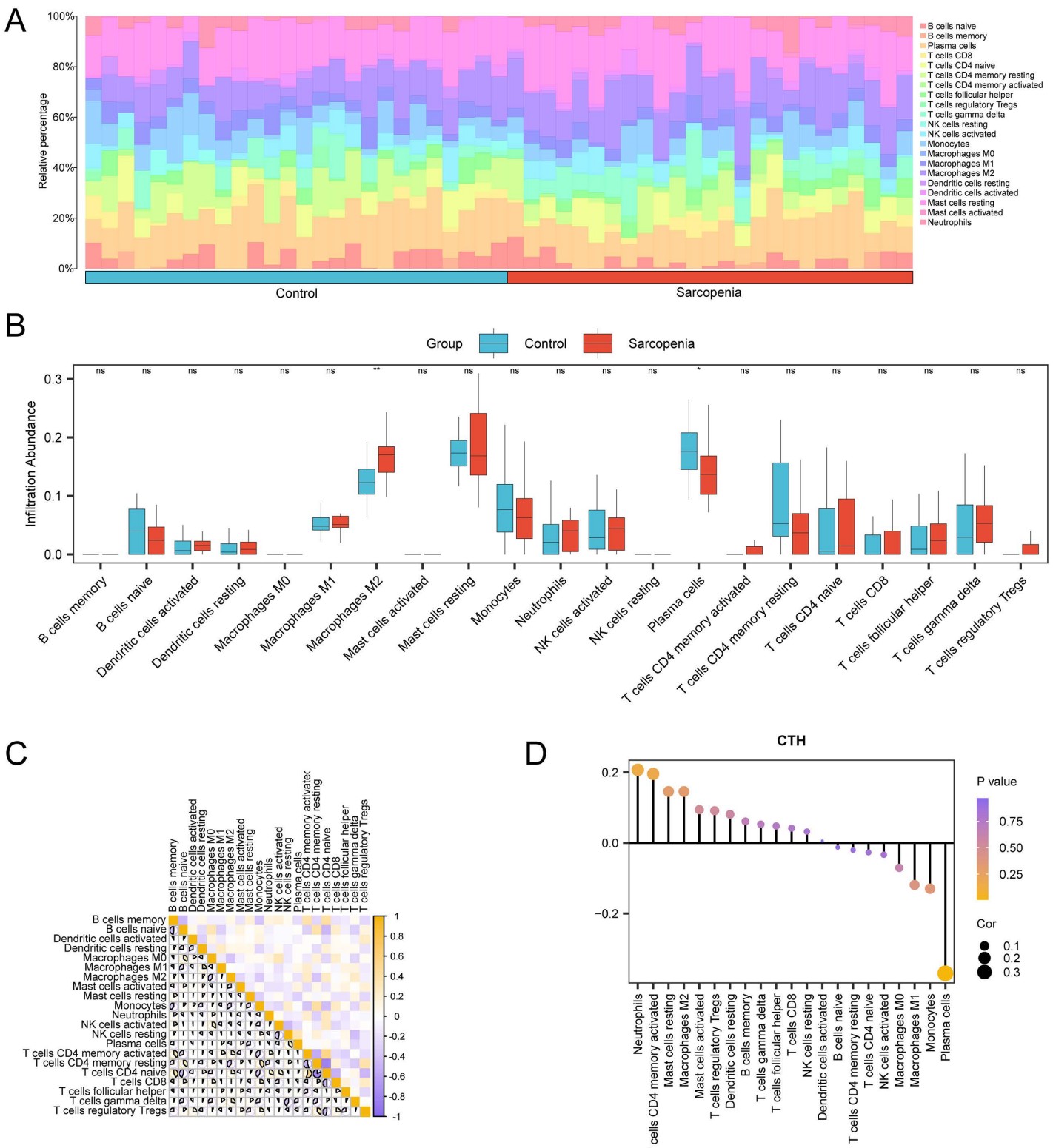

**Fig 15. Immune infiltration analysis by CIBERSORT algorithm.** A-b. Bar graph (A) and group comparison graph (B) of the proportion of immune cells in dataset GSE1428. C. Correlation heat map of immune cells in dataset GSE1428. D. Lollipop map of immune cell infiltration abundance and Model Genes in dataset GSE1428. The absolute value of the correlation coefficient (r-value) below 0.3 indicates weak or no correlation, between 0.3 and 0.5

indicates weak correlation, between 0.5 and 0.8 indicates moderate correlation, and above 0.8 indicates strong correlation. Light blue was a Control group and red was a Sarcopenia group. Yellow was a positive correlation, and vine purple was a negative correlation. The depth of the color indicates the strength of the correlation.

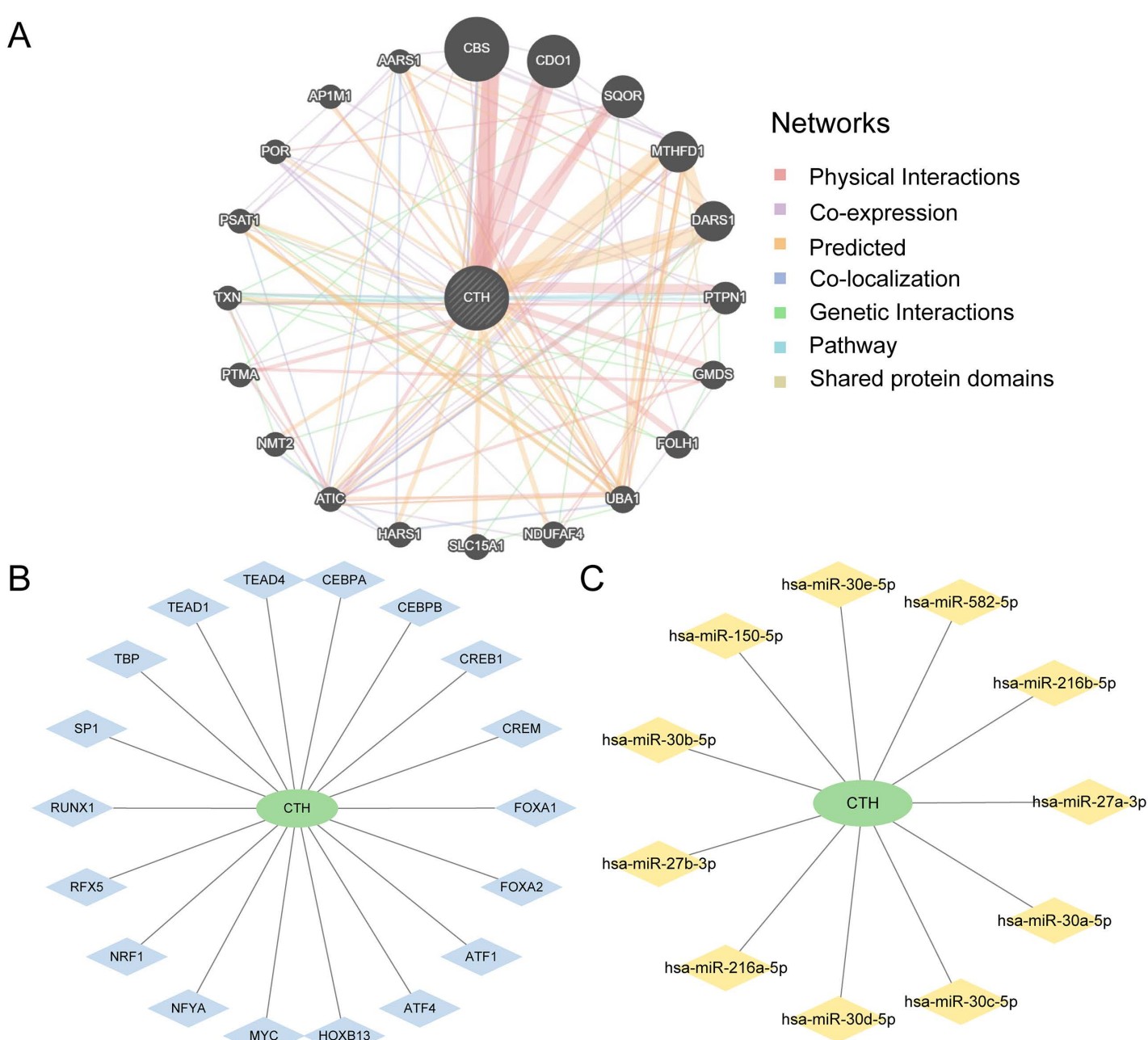

**Fig 16. Network of crosstalk genes.** A. The GeneMANIA website predicts the interaction network of functionally similar Genes of Model Genes. B. The mRNA-TF Regulatory Network of Model Genes. C. The mRNA-miRNA Regulatory Network of Model Genes. TF, Transcription Factor. Green is mRNA, blue is transcription factor TF, yellow is miRNA.

during the aging process and its disability can result in the older person becoming frail and less self-sufficient [35]. Since both IPF and sarcopenia are related to aging and are defined by dysregulated pathways, studies must determine the relationship between these two diseases and potential treatments. This research examines the correlation between IPF and sarcopenia through the determination of common genes predisposing to ERSRCGs. Therefore, using sophisticated molecular biology approaches within the current context investigation, we wanted to determine the underlying pathways that may connect these two diseases. Based on the current study, we found thirteen ERSRCGs, which can largely affect the advancement of both diseases and thus may serve as therapeutic targets and diagnostic markers. The expression changes of some ERSRCGs in two diseases are not completely consistent, suggesting disease-specific differences that may arise from tissue origin and pathological specificity. GSEA/GSVA analysis shows that the pathways are relatively broad, some of which have limited direct relationships with ER stress and may indirectly affect disease progression through processes such as metabolism, inflammation, or fibrosis. This complexity suggests that the results need to be interpreted with caution [36]. The subsequent sessions will further discuss these outcomes in biological contexts concerning differential gene expression, gene regulatory networks, and immune cell infiltration. They may define a new paradigm in the management of the affected patients & disease processes which are hitherto very difficult to manage.

Identifying these 13 ERSRCGs including *FOXO1*, *KAT2A*, and *CTH* brings attention to the central contribution of ER stress in the development of IPF and sarcopenia. Conserved as a transcription factor, *FOXO1* plays an essential part in the control of numerous metabolic processes and cell stature responses to stress. Its ability to modulate antioxidant processes and apoptosis-related pathways underlines that a strategy targeting this protein could be a preferable approach for enhancing cellular protection against damage caused by ER stress [37]. Furthermore, it encoded product *KAT2A* which is implicated in chromatin remodelling and gene regulation in stress response pathways, as a histone acetyltransferase [38]. Recent studies have shown how modifying the functional relationships of these genes can give positive cellular outcomes in models with ER stress. For instance, the activation of *FOXO1* can increase the expression of stress-protective genes. At the same time, the role of *KAT2A* in histone acetylation might help up-regulate chaperone proteins that are important in proper protein folding and degradation [39,40]. In addition, there is a synergism, which is proven by the finding that the interaction between *FOXO1* and *KAT2A* optimises the protective function against ER stress. Understanding more precisely the nature and specifics of the processes that govern these connexions and the relevance of these findings to therapeutic practise should be the points of focus of further research. Also, further investigation of whether these ERSRCGs could be used as biomarkers of IPF and sarcopenia can improve the diagnostic accuracy and promote early intervention. As a result, details of future studies addressing the detection and pharmacological management of agents interacting with the ER stress should be a priority.

In general, the validity of the diagnostic models was evident with the values of the predictive models of IPF; *CTH*, and *IDI1* as well as the models of sarcopenia; *FOFOXO1, CTH, HSD11B, GSTK1*, and SPTSSA. *CTH* (Cystathionine gamma-lyase) is essential for the metabolism of amino acids that contain sulfur and contributes to the creation of hydrogen sulfide, a signaling molecule with various physiological functions. Ischemic postconditioning in the context of IPF enhances *CTH* expression, which promotes inflammation and fibrogenesis and can perhaps, be used as a biomarker of the degree of disease progression [41]. But yet, *IDI1* is associated with the mevalonate pathway through which isoprenoids are synthesised. These compounds are involved in many processes which occurs within the cell and quality of the cell membranes. In light of the recent studies, alteration in the expression of *IDI1* has the potential to influence tumor progression and metastasis, and may well provide a hopeful avenue for therapies [42]. Therefore, the combination of the expression profiles of *CTH* and *IDI1* in clinical decision-making could improve personalized treatment strategies for patients. For instance, patients with high *CTH* levels might be anticipated to respond to treatments that influence oxidative stress, or patients with changes in *IDI1* may be appropriate for treatments involving metabolic pathways [43]. In view of the sarcopenic situation, *FOXO1* (Forkhead box O) is a transcription factor for muscle atrophy and metabolism; the promotion of which is very essential for the preservation of muscles [44]. *HSD11B1* is known to be involved in the

activation of cortisone to cortisol, which in turn regulates metabolisms and inflammatory response [45], *GSTK1* is known to participate in phase II detoxification modifying the cellular redox state [46]. *SPTSSA* is involved in an enzyme complex that synthesizes sphingolipids that are important in cellular communication and membrane stability [47]. These findings suggest that the roles of these genes in IPF disease alerts or prognostications should be the focus of future research for a further understanding of the resulting clinical characteristics.

Out of all immune cells, we distinguished 16 immune subtypes, which were upregulated in IPF samples as well as 21 subtypes identified in sarcopenia. Using this variation, they have highlighted that different disease states conform to distinct immune-related niches, which implies that likely distinct immune cell types may help to either exacerbate mist inflammation or facilitate tissue repair [48]. In the IPF samples, it was found that the majority of the immune cells have a high positive correlation, and maximum positive correlation between the Mast cells resting state and active NK cells was found. Mast cells can release mediators such as histamine and TGF-β, promoting the activation of fibroblasts and collagen deposition, thereby exacerbating the fibrotic process; activated NK cells, through cytotoxic effects and IFN-γ secretion, may, to some extent eliminate abnormal fibroblasts and exert anti-fibrotic effects. This significant positive correlation may reflect the dynamic balance between "pro-fibrotic and anti-fibrotic" forces during the fibrotic process, helping to explain the duality of immune response in IPF. In sarcopenia samples, the significant negative correlation between resting mast cells and activated mast cells suggests a balancing role for these cells in the processes of inflammation and repair. Mast cells not only participate in local inflammatory responses but also play a regulatory role in muscle regeneration. If the dynamic balance between resting and activated states is disrupted, it may lead to a persistent inflammatory environment, disrupting muscle homeostasis and accelerating muscle atrophy and functional decline. This finding provides a new perspective for understanding the immunological basis of sarcopenia [49]. Furthermore, the correlation between model genes and immune cells also reveals a potential "metabolic genes - immune cells - disease phenotype" connection. In IPF, *CTH* is significantly negatively correlated with γδT cells. γδT cells are usually involved in immune monitoring and inflammatory response, while CTH may affect immune cell activation through its metabolites (such as H2S), thereby inhibiting γδT cell function and altering the local inflammatory environment. In sarcopenia, *CTH* is negatively correlated with eosinophils. Eosinophils can secrete various cell factors to regulate muscle inflammation and regeneration environment, and the downregulation of *CTH* may weaken their recruitment or activity, thus affecting muscle homeostasis. The above results suggest that key ERSRCGs may participate in the pathological processes of IPF and sarcopenia by regulating immune cell function [50].

In our study, we built a thorough regulatory network and PPI network, identifying *CTH* and 20 functionally similar proteins, including CBS (cystathionine beta-synthase) and SQOR (sulfide quinone oxidoreductase). This PPI network provided some clue on the potential crosstalk between these genes in regulating fibrotic processes and muscle atrophy, and CBS being identified to be related to H2S generation and cardiovascular disease, we can imagine it would have a more extensive effect on tissue remodeling and repair in IPF patients [51]. The regulatory network included *CTH* and 18 transcription factors such as MYC and SP1, which are recognized for their roles in cellular stress responses and inflammation [52]. The involvement of these TFs suggests a mechanism by which they may mediate the expression of genes implicated in fibrosis and muscle metabolism, thus emphasizing *CTH*'s critical role in these pathological processes.The mRNA-miRNA regulation network also connected 11 related microRNAs encompassing *hsa-miR-27a-3p* and *hsa-miR-30a-5p* to *CTH*. These observations confirm the complex interplay of such regulation under inflammatory conditions and raise the possibility of differential post-transcriptional regulation of *CTH* and its pathways as critical in the development of IPF and sarcopenia.

The suboptimal performance of our diagnostic model in the independent validation cohort (AUC ≈ 0.56) underscores the difficulty in translating bioinformatic findings into reliable clinical tools. Key contributing factors include the significant disease heterogeneity of both IPF and sarcopenia, as well as technical variations such as batch effects between cohorts. Despite its limited diagnostic utility, the model may serve as a cost-effective tool for preliminary sarcopenia

risk screening in IPF patients, helping prioritize individuals for more definitive assessments. Moreover, it offers valuable insights into shared molecular mechanisms between IPF and sarcopenia, supporting future research into therapeutic targets.

In this study, a threshold of adj.p < 0.05 and |logFC| > 0 was used during the screening of differentially expressed genes. This standard is relatively lenient, and while it helps to capture potential molecular signals as much as possible in exploratory analyses, it inevitably increases the risk of false positive results. Therefore, future validation of the robustness and biological functions of these potential candidate genes needs to be conducted through data analysis and experimental studies with larger sample sizes. This study relied solely on public transcriptome databases for bioinformatic analysis and did not include validation via in vivo or in vitro experiments. Although potential hub genes and diagnostic models were identified using computational methods, these findings require experimental confirmation. Future work should focus on elucidating the functions of key genes (e.g., *CTH* and *FOXO1*) through cellular assays, animal models, and clinical samples to verify their diagnostic and prognostic value. These efforts will help clarify the molecular mechanisms linking IPF and sarcopenia and facilitate clinical translation.

## 5 Conclusion

To summarize, this study offers a clear understanding of the molecular pathways shared by both diseases by extracting 13 genes with potential connexions to the two diseases at multiple levels of data analysis. These results therefore offer novel knowledge that may be a foundation for prospective diagnostic techniques or specific therapies to improve patient experience. Future research is needed to replicate these findings on diverse clinical samples with larger formal clinical patient populations and to conatively assess the roles of those identified genes.

## Supporting information

**S1 File. Supplementary Tables S1–S7**
(ZIP)

## Acknowledgments

We thank the public databases for supplying the necessary data for our study. Additionally, we acknowledge the developers of R software and its packages for their valuable contributions and the convenience they provided in our research.

## Author contributions

**Data curation:** Lanying Shen, Zihan Yi.

**Formal analysis:** Lanying Shen, Zihan Yi.

**Methodology:** Lanying Shen, Jiahao Liu.

**Software:** Jiahao Liu.

**Supervision:** Yinghua Ying.

**Validation:** Yinghua Ying.

**Writing – original draft:** Lanying Shen.

**Writing – review & editing:** Lanying Shen, Yue Hu.

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
