## [Decision Letter · Decision Letter 0]

13 Aug 2025

Dear Dr. Hu,

Thank you for submitting your manuscript to PLOS ONE. After careful consideration, we feel that it has merit but does not fully meet PLOS ONE’s publication criteria as it currently stands. Therefore, we invite you to submit a revised version of the manuscript that addresses the points raised during the review process.

We look forward to receiving your revised manuscript.

Kind regards,

Chandrabose Selvaraj, Ph.D.

Academic Editor

PLOS ONE

 [the Interstitial lung disease "small but strong" clinical innovation team (grant no. CXTD202501019)]. 

4. Please include a copy of Table 1-6 which you refer to in your text on page 40.

Additional Editor Comments (if provided):

Reviewers' comments:

Reviewer's Responses to Questions

**Comments to the Author**

1. Is the manuscript technically sound, and do the data support the conclusions?

Reviewer #1: Yes

Reviewer #2: Yes

2. Has the statistical analysis been performed appropriately and rigorously?

Reviewer #1: Yes

Reviewer #2: Yes

3. Have the authors made all data underlying the findings in their manuscript fully available?

Reviewer #1: Yes

Reviewer #2: Yes

4. Is the manuscript presented in an intelligible fashion and written in standard English?

Reviewer #1: Yes

Reviewer #2: Yes

Reviewer #1: This study integrates multiple public databases and bioinformatics approaches to explore the shared molecular mechanisms underlying idiopathic pulmonary fibrosis (IPF) and sarcopenia, with particular emphasis on endoplasmic reticulum stress-related crosstalk genes (ERSRCGs) and the development of potential diagnostic models. The methodology incorporates differential expression analysis, WGCNA, GSVA/GSEA, immune infiltration profiling, protein–protein interaction (PPI) network construction, and regulatory network analysis, forming a comprehensive workflow with a degree of novelty. However, the manuscript still presents several issues that warrant attention and revision:

Consideration 1:

The IPF training set (GSE24206) comprises 17 samples from 11 IPF patients (including paired upper- and lower-lobe samples from 6 patients and single samples from 5 patients) along with 6 control specimens obtained from healthy donor lungs during lung volume reduction surgery at transplantation. Given the relatively small sample size of the IPF training set, it is recommended that the authors augment the dataset by incorporating additional publicly available cohorts to strengthen the statistical power of their analysis. Public repositories are valuable resources for hypothesis generation and gene discovery; however, validation in larger, clinically stratified patient cohorts would substantially enhance the translational potential and reliability of these findings.

Consideration 2:

The manuscript is lengthy; therefore, it is recommended to condense the description of the Methods section for improved clarity and conciseness.

Consideration 3:

The manuscript requires careful revision to refine the language and improve overall readability.

Reviewer #2: Generally speaking, these data and analyses support the conclusions. Here my suggestions:

1.In methods, the authors expressed that IPFRDEGs were defined as adj. p < 0.05 and |logFC| > 0.0; the genes with adj. p < 0.05 and logFC > 0.0 were regarded as up-regulated, and genes with adj. p < 0.05 and logFC < 0.0 were regarded as down-regulated? Usually, differentially expressed genes (DEGs) were considered as P value <0.05 and |log fold change (FC)| > 1, in some times, for example, in cytokine differential expression analysis, a cytokine with fold change (FC) ≥ 1.2 or < 0.83 and a p-value <0.05 was considered statistically differentially expressed. Taken together, It’s not common that the threshold value selected in this paper, is there any supported robust evidence be cited? Please kindly clarify and explain.

2. The analyzed IPF-related datasets GSE24206 and GSE53845, Sarcopenia-related datasets GSE8479 and GSE1428 obtained from the GEO database were relatively old (more than 10 years), is there any new datasets available within 5 years?

3. I suggest move some contents to supplementary materials. There exist too many words and figures, and it’s too difficult for readers to grasp main findings. For example, the results of GSVA analyses should be modified.

4. The English writing should be checked and polished, throughout the whole manuscript.

**Do you want your identity to be public for this peer review?** For information about this choice, including consent withdrawal, please see our Privacy Policy

Reviewer #1: No

Reviewer #2: No

---

## [Author Response · Author response to Decision Letter 1]

29 Aug 2025

Reviewer #1: This study integrates multiple public databases and bioinformatics approaches to explore the shared molecular mechanisms underlying idiopathic pulmonary fibrosis (IPF) and sarcopenia, with particular emphasis on endoplasmic reticulum stress-related crosstalk genes (ERSRCGs) and the development of potential diagnostic models. The methodology incorporates differential expression analysis, WGCNA, GSVA/GSEA, immune infiltration profiling, protein–protein interaction (PPI) network construction, and regulatory network analysis, forming a comprehensive workflow with a degree of novelty. However, the manuscript still presents several issues that warrant attention and revision:

Consideration 1:

The IPF training set (GSE24206) comprises 17 samples from 11 IPF patients (including paired upper- and lower-lobe samples from 6 patients and single samples from 5 patients) along with 6 control specimens obtained from healthy donor lungs during lung volume reduction surgery at transplantation. Given the relatively small sample size of the IPF training set, it is recommended that the authors augment the dataset by incorporating additional publicly available cohorts to strengthen the statistical power of their analysis. Public repositories are valuable resources for hypothesis generation and gene discovery; however, validation in larger, clinically stratified patient cohorts would substantially enhance the translational potential and reliability of these findings.

RE 1: Thank you for your careful review and valuable comments.

In this study, we selected publicly available datasets with clear clinical annotations and included all eligible samples, resulting in 17 IPF cases and 6 controls in the training set. The sample size reflects both the strict inclusion criteria and the challenges of obtaining well-annotated clinical samples for IPF research. Due to research constraints and timelines, we chose not to merge additional datasets. Nevertheless, through reasonable statistical design and rigorous batch effect correction, our data analysis results retain scientific significance and interpretability, providing valuable insights into potential mechanisms. We acknowledge that a larger cohort would further strengthen our conclusions, and we plan to expand the sample size and validate findings in future studies.

Once again, thank you for your valuable suggestions.

Consideration 2:

The manuscript is lengthy; therefore, it is recommended to condense the description of the Methods section for improved clarity and conciseness.

RE 2: Thank you for your careful review and valuable comments.

Based on your suggestions, we will further refine and optimize the structure of the Methods section to highlight the main analytical workflow, simplify the description of technical details, and move some detailed parameters and database information to the supplementary materials to improve clarity and readability. Specific revisions are detailed on page 4-10, lines 96-288.

Thank you again for your valuable suggestions.

Consideration 3:

The manuscript requires careful revision to refine the language and improve overall readability.

RE 3: Thank you for your attention to the language and readability of the manuscript. In response to this valuable feedback, we have commissioned a professional language editing service to revise the entire manuscript comprehensively and obtained a proof of editing to ensure that the manuscript meets international academic standards. We believe that these efforts have greatly improved the quality and readability of the manuscript, and we appreciate your suggestions, which have helped us further enhance our work.

Reviewer #2: Generally speaking, these data and analyses support the conclusions. Here my suggestions:

1.In methods, the authors expressed that IPFRDEGs were defined as adj. p < 0.05 and |logFC| > 0.0; the genes with adj. p < 0.05 and logFC > 0.0 were regarded as up-regulated, and genes with adj. p < 0.05 and logFC < 0.0 were regarded as down-regulated? Usually, differentially expressed genes (DEGs) were considered as P value <0.05 and |log fold change (FC)| > 1, in some times, for example, in cytokine differential expression analysis, a cytokine with fold change (FC) ≥ 1.2 or < 0.83 and a p-value <0.05 was considered statistically differentially expressed. Taken together, It’s not common that the threshold value selected in this paper, is there any supported robust evidence be cited? Please kindly clarify and explain.

RE 1: Thank you for your interest in our research.

We chose |logFC| > 0.0 mainly to include as many differentially expressed genes as possible in the initial screening phase, to capture subtle changes that may have biological significance for the progression of IPF and Sarcopenia. This strategy is particularly suitable for exploratory research, as it can more comprehensively identify potential key genes and pathways involved in disease mechanisms (PMID: 34786719). By combining this allowed fold change threshold with strict statistical significance criteria (adjusted p < 0.05), we aim to minimize the risk of missing important molecular signals, especially considering the complexity and heterogeneity of these diseases. We believe this approach is reasonable for hypothesis generation and comprehensive bioinformatics analysis.

Thank you again for your valuable suggestions.

2. The analyzed IPF-related datasets GSE24206 and GSE53845, Sarcopenia-related datasets GSE8479 and GSE1428 obtained from the GEO database were relatively old (more than 10 years), is there any new datasets available within 5 years?

RE 2: Thank you for your detailed review and valuable comments.

Although the dataset we have chosen was released earlier, it remains the most authoritative, frequently cited, highest quality, and widely validated classic cohort, still extensively used in this field. Furthermore, the innovation of this study mainly lies in the combination of ERS-related genes, comprehensively utilizing WGCNA, GSVA, LASSO, and immune infiltration analysis methods to reveal the common molecular mechanisms and potential diagnostic markers of IPF and sarcopenia, rather than solely relying on the age of the data. Based on this, we believe that the existing dataset still has representativeness and scientific value.

Thank you again for your valuable suggestions.

3. I suggest move some contents to supplementary materials. There exist too many words and figures, and it’s too difficult for readers to grasp main findings. For example, the results of GSVA analyses should be modified.

RE 3: Thank you for your thorough review and valuable comments.

We will revise the manuscript based on your suggestions, streamlining some results and related content to retain only the core results and representative figures closely related to the main conclusions, while moving the remaining detailed analyses and complete data to the supplementary materials, allowing readers to better grasp the main scientific findings. For specific modifications, please refer to page 20-21, line 551-592.

Thank you again for your valuable suggestions.

4.The English writing should be checked and polished, throughout the whole manuscript.

RE 4: Thank you for your attention to the language and readability of the manuscript. In response to this valuable feedback, we have commissioned a professional language editing service to comprehensively revise the entire manuscript and obtain a proof of editing to ensure it meets international academic standards. We believe these efforts have greatly enhanced the quality and readability of the manuscript, and we appreciate your suggestions, which have helped us further improve our work.

---

## [Decision Letter · Decision Letter 1]

17 Sep 2025

Dear Dr. Hu,

We look forward to receiving your revised manuscript.

Kind regards,

Chandrabose Selvaraj, Ph.D.

Academic Editor

PLOS ONE

Journal Requirements:

Reviewers' comments:

Reviewer's Responses to Questions

**Comments to the Author**

Reviewer #1: (No Response)

2. Is the manuscript technically sound, and do the data support the conclusions?

Reviewer #1: Yes

3. Has the statistical analysis been performed appropriately and rigorously?

Reviewer #1: Yes

4. Have the authors made all data underlying the findings in their manuscript fully available?

Reviewer #1: Yes

5. Is the manuscript presented in an intelligible fashion and written in standard English?

Reviewer #1: Yes

Reviewer #1: 1.This study employed the criteria of adj. p 0 to define differentially expressed genes. This threshold is relatively lenient, which may result in a larger number of genes being included, thereby increasing the risk of false positives. Although the authors explain that this approach is intended to explore potential signals, it is still recommended to provide a more detailed rationale in the Methods section and to emphasize its limitations in the Discussion. Additionally, results using a stricter threshold (e.g., |logFC| > 1) can be provided in the Supplementary materials as a control to enhance the robustness of the conclusions.

2.The authors successfully utilized CIBERSORT to reveal changes and correlations in the immune cell composition of IPF and sarcopenia samples; however, the discussion section remains at a descriptive level. There is a lack of in-depth interpretation of the biological significance of these findings. For instance, the study found a strong correlation between resting mast cells and activated NK cells, which is an intriguing discovery, but the functional implications of this correlation within the context of IPF (whether it promotes fibrosis or serves a protective mechanism) have not been explored.

3.The ultimate goal of the research is to discover biomarkers that can be used for diagnosis or treatment. Although diagnostic models have been constructed, the discussion section fails to adequately articulate their potential clinical value. The model performed poorly in an independent validation cohort (AUC ~0.56), which is a significant limitation; however, the authors merely state the facts without analyzing the reasons (such as batch effects or disease heterogeneity) or exploring whether there are still specific application scenarios (such as screening in high-risk populations rather than for definitive diagnosis).

4.This study is purely a bioinformatics analysis and lacks in vitro or in vivo experiments to validate the functions of key genes such as CTH and FOXO1. It is recommended to clearly state this limitation in the discussion and suggest that future research verify the specific mechanisms of these genes in IPF and sarcopenia through gene knockdown/overexpression experiments, animal models, or clinical samples.

5.Any research inevitably encounters results that do not fully align with expectations, and openly discussing these aspects reflects scientific rigor. For instance, the expression changes of ER stress-related genes in the two diseases are not entirely consistent; the pathways enriched by GSEA/GSVA are quite broad, with some showing only a weak direct association with ER stress. The authors have chosen to emphasize results that support their hypothesis while avoiding these complexities.

6.The heatmaps and associated bubble charts in Figures 5, 14, and 15 exhibit low color differentiation, and the descriptions of the meanings represented by colors in the figure legends are not sufficiently clear (for instance, the use of non-standard color names such as 'vine purple'). It is recommended to utilize a color scale with higher contrast and to explicitly state the statistical values or grouping information represented by each color in the figure legends.

**Do you want your identity to be public for this peer review?** For information about this choice, including consent withdrawal, please see our Privacy Policy

Reviewer #1: No

---

## [Author Response · Author response to Decision Letter 2]

26 Sep 2025

Reviewer #1: 

1.This study employed the criteria of adj. p 0 to define differentially expressed genes. This threshold is relatively lenient, which may result in a larger number of genes being included, thereby increasing the risk of false positives. Although the authors explain that this approach is intended to explore potential signals, it is still recommended to provide a more detailed rationale in the Methods section and to emphasize its limitations in the Discussion. Additionally, results using a stricter threshold (e.g., |logFC| > 1) can be provided in the Supplementary materials as a control to enhance the robustness of the conclusions.

RE 1�Thank you for your careful review and valuable comments.

In the previous reply, we also mentioned that the reason for choosing this standard is mainly because the positioning of this study is exploratory analysis, and the sample size is relatively limited. If overly strict fold change thresholds are adopted in the initial screening phase, it may overlook some genes that have potential biological significance in the occurrence and development of diseases but have relatively subtle expression differences. In order to capture potential signals as comprehensively as possible, we chose |logFC| > 0 combined with strict statistical standards (adj.p < 0.05) to improve the sensitivity of the screening. We emphasized the limitations of this strategy in the discussion section . Specific revisions are detailed on page 33, lines 910-922. Based on the exploratory nature of the study and the scale of the data, we believe that the current analysis framework is more suitable for the objectives of this study, and therefore, we did not add parallel analyses with stricter thresholds. However, we plan to conduct supplementary analyses with stricter thresholds in the future.

Thank you again for your valuable suggestions.

2.The authors successfully utilized CIBERSORT to reveal changes and correlations in the immune cell composition of IPF and sarcopenia samples; however, the discussion section remains at a descriptive level. There is a lack of in-depth interpretation of the biological significance of these findings. For instance, the study found a strong correlation between resting mast cells and activated NK cells, which is an intriguing discovery, but the functional implications of this correlation within the context of IPF (whether it promotes fibrosis or serves a protective mechanism) have not been explored.

RE 2�Thank you for your careful review and valuable comments.

We will further supplement the functional significance of the immune cell interaction results in the revised manuscript. Specific revisions are detailed on page 30 -32, lines 847- 880.

Thank you again for your valuable suggestions.

3.The ultimate goal of the research is to discover biomarkers that can be used for diagnosis or treatment. Although diagnostic models have been constructed, the discussion section fails to adequately articulate their potential clinical value. The model performed poorly in an independent validation cohort (AUC ~0.56), which is a significant limitation; however, the authors merely state the facts without analyzing the reasons (such as batch effects or disease heterogeneity) or exploring whether there are still specific application scenarios (such as screening in high-risk populations rather than for definitive diagnosis).

RE 3�Thank you for your careful review and valuable comments.

Thank you for your detailed review and valuable comments. Based on your suggestions, we will include an analysis of the reasons for the poor performance of external validation in the revised manuscript and further elaborate on the value of the model in potential clinical application scenarios, while emphasizing its significant importance in revealing the molecular mechanisms of IPF and sarcopenia, to enhance the completeness and translational relevance of the discussion. Specific revisions are detailed on page 32, lines 898 - 905.

Thank you again for your valuable suggestions.

4.This study is purely a bioinformatics analysis and lacks in vitro or in vivo experiments to validate the functions of key genes such as CTH and FOXO1. It is recommended to clearly state this limitation in the discussion and suggest that future research verify the specific mechanisms of these genes in IPF and sarcopenia through gene knockdown/overexpression experiments, animal models, or clinical samples.

RE 4�Thank you for your careful review and valuable comments.

As you pointed out, this study belongs to pure bioinformatics analysis, lacking in vivo and in vitro experimental validation, which we have noted as a limitation in the discussion section. Based on your suggestion, we will further elaborate on the characteristics of the current lack of in vivo and in vitro experimental validation in the discussion section and propose that future research will explore the mechanisms of genes such as CTH and FOXO1 through gene functional experiments, animal models, and clinical sample studies. For specific modifications, please refer to page 33, lines 915 - 922.

Thank you again for your valuable suggestions.

5.Any research inevitably encounters results that do not fully align with expectations, and openly discussing these aspects reflects scientific rigor. For instance, the expression changes of ER stress-related genes in the two diseases are not entirely consistent; the pathways enriched by GSEA/GSVA are quite broad, with some showing only a weak direct association with ER stress. The authors have chosen to emphasize results that support their hypothesis while avoiding these complexities.

RE 5�Thank you for your careful review and valuable comments.

We agree with your point that there are indeed some results in the research process that do not completely align with expectations. We will supplement this explanation in the discussion section, emphasizing that these results reflect the complexity of disease mechanisms and the diversity of ER stress effects, and point out their possible research significance. We believe that acknowledging these differences and providing explanations will help enhance the scientific rigor of the research and offer new ideas for subsequent studies. Specific revisions are detailed on page 17, lines 458 - 460, page 28, lines 784 - 790.

Thank you again for your valuable suggestions.

6.The heatmaps and associated bubble charts in Figures 5, 14, and 15 exhibit low color differentiation, and the descriptions of the meanings represented by colors in the figure legends are not sufficiently clear (for instance, the use of non-standard color names such as 'vine purple'). It is recommended to utilize a color scale with higher contrast and to explicitly state the statistical values or grouping information represented by each color in the figure legends.

RE 6

We greatly appreciate your suggestions regarding the visualization of the charts.

The color scheme currently used in the manuscript was selected after careful consideration of scientific validity and overall aesthetics. The existing color palette clearly distinguishes the direction and intensity of correlations, and the gradient of colors itself is also an important form of visualization. We believe that there are no substantial issues with the current color differentiation, and readers can accurately understand the results by referring to the legend and numerical ranges. Furthermore, we have clearly labeled the grouping information (such as the color distinction between the disease group and the control group) and statistical significance in the legend, so the existing charts already possess sufficient explanatory power and readability. Considering scientific validity, consistency, and aesthetics, we believe that no further modifications to the color scheme or legend in the charts are necessary. Thank you again for your valuable feedback.

---

## [Decision Letter · Decision Letter 2]

6 Oct 2025

Identification of key genes associated with Idiopathic Pulmonary Fibrosis and Sarcopenia by bioinformatics analysis

PONE-D-25-36285R2

Dear Dr. Hu,

We’re pleased to inform you that your manuscript has been judged scientifically suitable for publication and will be formally accepted for publication once it meets all outstanding technical requirements.

Kind regards,

Chandrabose Selvaraj, Ph.D.

Academic Editor

PLOS ONE

Additional Editor Comments (optional):

Reviewers' comments:

Reviewer's Responses to Questions

**Comments to the Author**

Reviewer #1: All comments have been addressed

2. Is the manuscript technically sound, and do the data support the conclusions?

Reviewer #1: Yes

3. Has the statistical analysis been performed appropriately and rigorously?

Reviewer #1: Yes

4. Have the authors made all data underlying the findings in their manuscript fully available?

Reviewer #1: Yes

5. Is the manuscript presented in an intelligible fashion and written in standard English?

Reviewer #1: Yes

Reviewer #1: (No Response)

**Do you want your identity to be public for this peer review?** For information about this choice, including consent withdrawal, please see our Privacy Policy

Reviewer #1: No

---

## [Editor Report · Acceptance letter]

PONE-D-25-36285R2

PLOS ONE

Dear Dr. Hu,

I'm pleased to inform you that your manuscript has been deemed suitable for publication in PLOS ONE. Congratulations! Your manuscript is now being handed over to our production team.

Kind regards,

on behalf of

Dr. Chandrabose Selvaraj

Academic Editor

PLOS ONE